# MITIGATING CATASTROPHIC FORGETTING IN TARGET LANGUAGE ADAPTATION OF LLMS VIA SOURCE-SHIELDED UPDATES

## ABSTRACT

Expanding the linguistic diversity of instruct large language models (LLMs) is crucial for global accessibility but is often hindered by the reliance on costly specialized target language labeled data and catastrophic forgetting during adaptation. We tackle this challenge under a realistic, low-resource constraint: adapting instruct LLMs using only unlabeled target language data. We introduce **S**ource-**S**hielded **U**pdates (**SSU**), a selective parameter update strategy that proactively preserves source knowledge. Using a small set of source data and a parameter importance scoring method, SSU identifies parameters critical to maintaining source abilities. It then applies a column-wise freezing strategy to protect these parameters before adaptation. Experiments across five typologically diverse languages and 7B and 13B models demonstrate that SSU successfully mitigates catastrophic forgetting. It reduces performance degradation on monolingual source tasks to just 3.4% (7B) and 2.8% (13B) on average, a stark contrast to the 20.3% and 22.3% from full fine-tuning. SSU also achieves target-language performance highly competitive with full fine-tuning, outperforming it on all benchmarks for 7B models and the majority for 13B models.[1]

## 1 INTRODUCTION

Large language models (LLMs) demonstrate remarkable generalization capabilities across numerous applications (OpenAI, 2025; DeepSeek-AI et al., 2025; Yang et al., 2025; Gemma Team et al., 2025). However, they notoriously underperform in languages absent or underrepresented in their training data, creating a critical barrier to equitable access for speakers worldwide (Huang et al., 2023). The standard approach to resolve this issue is to continue pre-training (CPT) or fine-tune on target language data, i.e., target language adaptation (Cui et al., 2024; Ji et al., 2025).

Yet, adapting instruct models to these languages is uniquely challenging. Such models require specialized instruction-tuning data (Wei et al., 2022; Rafailov et al., 2023), which is often unavailable or prohibitively costly to create for underrepresented languages (Huang et al., 2024c). Furthermore, machine-translated data as a low-cost alternative is not consistently effective (Tao et al., 2024).

Consequently, unlabeled target language text is often the only viable data for adaptation. While this approach can improve target language proficiency, it often triggers catastrophic forgetting (Kirkpatrick et al., 2017; Tejaswi et al., 2024; Mundra et al., 2024; Yamaguchi et al., 2025), where new training erases prior knowledge. This issue is particularly acute for instruct models, as it cripples the general-purpose functionality of the model, which is primarily derived from core abilities like chat and instruction-following. In response, previous work has attempted **post-hoc** mitigation. For example, Yamaguchi et al. (2025) merge the weights of the original instruct model with the corresponding adapted model, while Huang et al. (2024c) treat adaptation as a task vector, applying parameter changes from CPT on the base model to the instruct model. Nonetheless, these methods largely fail to mitigate catastrophic forgetting, substantially degrading these core functionalities.

The shortcomings of post-hoc methods suggest that *mitigation should occur during adaptation.* We therefore turn our focus to **the CPT stage**. Specifically, we leverage selective parameter updates,

---

[1]Our anonymous code is available on https://anonymous.4open.science/r/ssu-iclr-2026/.

a method of restricting which weights are modified during training. This approach is proven more effective at mitigating catastrophic forgetting than alternatives like parameter-efficient fine-tuning, regularization, or model merging (Zhang et al., 2024a; Hui et al., 2025). However, existing selective parameter tuning paradigms for adapting LLMs are ill-suited for the specific challenge of adapting instruct models with unlabeled target language text. They either rely on **random selection**, offering no principled way to preserve knowledge, or on signals from the new data to guide updates (**target-focused**). The latter approaches are particularly vulnerable in this scenario because signals from raw, target unstructured text are misaligned with the core chat and instruction-following capabilities of the models. Optimizing for this out-of-distribution format risks corrupting the very foundational capabilities we aim to preserve.

We therefore introduce **S**ource-**S**hielded **U**pdates (**SSU**), a novel **source-focused** approach that *proactively shields source knowledge before adaptation begins* (Figure 1). First, SSU identifies parameters critical to source abilities using a small set of source data and a parameter importance scoring method, such as those used in model pruning (e.g., Wanda (Sun et al., 2024)). Second, it uses these element-wise scores to construct a column-wise freezing mask. This structural design is crucial. Unlike naive element-wise freezing that corrupts feature transformations, our column-wise approach preserves them entirely. Finally, this mask is applied during CPT on unlabeled target language data, keeping the shielded structural units frozen. This process allows SSU to effectively preserve the general-purpose ability of the model while improving target language performance.

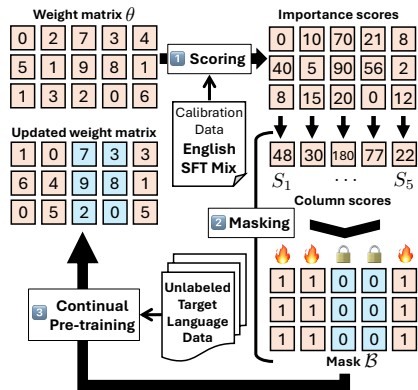

Figure 1: Overview of **S**ource-**S**hielded Update (**SSU**). The method comprises three stages: importance scoring, column-wise mask generation, and continual pre-training on unlabeled target language data with the masks.

We verify the effectiveness of our approach through extensive experiments with five typologically diverse languages and two different model scales (7B and 13B). We evaluate performance on the source language (English) across multiple dimensions, including chat and instruction-following, safety, and general generation and classification abilities, alongside performance on the target language. We summarize our contributions as follows:

- A novel method for adapting instruct models to a target language without specialized target instruction-tuning data, addressing a key bottleneck to expand linguistic accessibility.

- At two model scales, SSU consistently outperforms all baselines on all core instruction-following and safety tasks. It achieves leading target-language proficiency rivaling full fine-tuning while almost perfectly preserving general source-language performance.

- Extensive analysis validates the efficacy of SSU, confirming the superiority of column-wise freezing and the importance of source data-driven parameter scoring. Qualitatively, we observe that SSU avoids the linguistic code-mixing that state-of-the-art methods suffer from, explaining its superior abilities across source chat and instruction-following tasks.

## 2 RELATED WORK

**Language Adaptation.** CPT on target language data is the standard method for adapting LLMs to target languages (Cui et al., 2024; Fujii et al., 2024; Da Dalt et al., 2024; Cahyawijaya et al., 2024; Nguyen et al., 2024; Yamaguchi et al., 2024; Nag et al., 2025; Ji et al., 2025, *inter alia.*). While effective, CPT often leads to substantial degradation of the original capabilities of a model (Tejaswi et al., 2024; Mundra et al., 2024; Yamaguchi et al., 2025), a phenomenon known as catastrophic forgetting. This trade-off presents a major obstacle, especially for instruct models where preserving core chat and instruction-following abilities is vital for their general-purpose functionality.[2]

---

[2]While some research addresses tokenization overfragmentation, where words are split into inefficiently small units, via vocabulary adaptation (Tejaswi et al., 2024; Mundra et al., 2024; Yamaguchi et al., 2025, *inter alia.*), we focus on catastrophic forgetting during **parameter updates** with a fixed architecture. We consider vocabulary adaptation orthogonal to our approach; combining it with SSU offers a promising avenue for future work.

**Catastrophic Forgetting.** Mitigating catastrophic forgetting is a long-standing challenge in continual learning. Proposed solutions generally fall into five categories: (1) **Regularization-based** methods add a penalty term to the loss function to discourage significant changes to weights deemed important for previous tasks (Kirkpatrick et al., 2017; Chen et al., 2020; Zhang et al., 2022, *inter alia.*). (2) **Replay-based** methods interleave old and new data (de Masson d'Autume et al., 2019; Rolnick et al., 2019; Huang et al., 2024b, *inter alia.*). (3) **Model merging** methods interpolate between original and fine-tuned models (Wortsman et al., 2022; Yadav et al., 2023; Yu et al., 2024; Huang et al., 2024a, *inter alia.*). (4) **Architecture-based** methods like LoRA (Hu et al., 2022) add and train new parameters while freezing the original model (Houlsby et al., 2019; Hu et al., 2022; Zhang et al., 2023, *inter alia.*). (5) **Selective parameter updates** restrict which existing weights are modified during training (Zhang et al., 2024a; Hui et al., 2025). Our work belongs to this category.[3]

Studies on multilingual CPT for LLMs similarly employ these strategies. Examples include mixing source (English) data (Category 2) (Zheng et al., 2024; Elhady et al., 2025), model merging (Category 3) (Alexandrov et al., 2024; Blevins et al., 2024), and concurrent work on architecture-based solutions (Category 4) (Owodunni & Kumar, 2025). Optimization strategies, such as controlling learning rates (Winata et al., 2023), are also utilized. These methods are largely orthogonal to our work. SSU, in contrast, focuses on **selective parameter updates** (Category 5), distinguished by a proactive, source-driven approach which we detail next.

**Selective Parameter Updates.** While often utilized for training efficiency (Liu et al., 2021; Lodha et al., 2023; Li et al., 2023a; Pan et al., 2024; Yang et al., 2024; Li et al., 2024; Ma et al., 2024; Li et al., 2025; He et al., 2025), selective parameter updates have also proven effective for mitigating catastrophic forgetting (Zhang et al., 2024a; Hui et al., 2025). These methods can be broadly categorized as **dynamic** or **static**. Dynamic approaches select a trainable parameter set that can change during training, based on random selection (Li et al., 2024; Pan et al., 2024) or target data signals like gradient magnitudes (Liu et al., 2021; Li et al., 2023a; Ma et al., 2024; Li et al., 2025). In contrast, static methods define a fixed trainable parameter set before training or during warm-up. This allows for straightforward integration with existing pipelines, enabling the combination of orthogonal mitigation methods like regularization and replay more easily. For example, a method closest to our work (Hui et al., 2025) randomly freezes half of the components within each transformer sub-layer (i.e., self-attention, feed-forward, and layernorm), while others are data-driven based on target data (Lodha et al., 2023; Zhang et al., 2024a; Panda et al., 2024; He et al., 2025).

**SSU: A Source-Focused Selective Parameter Update Approach.** SSU is a static selective parameter update approach (Category 5) that introduces a new, source-focused paradigm for language adaptation. Unlike existing selective parameter update methods that rely on random choice or target data signals, SSU uses a small sample of source data (e.g., 500 samples) to identify and freeze parameters critical to the source knowledge within the model before adaptation. This also distinguishes it from previous importance-based methods in other categories. For instance, regularization methods (Category 1) are reactive, applying a penalty to weight changes (Jung et al., 2020). In contrast, SSU is proactive, using a static structural mask to prevent updates before adaptation. Similarly, SSU is not an architecture-based PEFT method (Category 4), which uses importance to insert new parameters (Yao et al., 2024). SSU instead operates on full, existing parameters to select and freeze structural columns.

## 3 SSU: Selective Parameter Updates via Importance Freezing

We address the challenge of adapting an instruct model using only raw, unlabeled target language data. Unlike prior work that focuses on post-hoc mitigation (Huang et al., 2024c; Yamaguchi et al., 2025), we introduce Source-Shielded Updates (SSU), a method that targets the CPT process itself. The goal is to mitigate catastrophic forgetting during CPT, thereby maintaining the general-purpose functionality of an instruct model. Concurrently, SSU aims to achieve performance gains in the target language tasks comparable to those from full fine-tuning. Formally, given an instruct model $\mathcal{M}$, calibration data $\mathcal{D}_{\text{calib}}$, unlabeled target language data $\mathcal{D}_{\text{target}}$, and a parameter freezing ratio $k$, SSU adapts $\mathcal{M}$ on $\mathcal{D}_{\text{target}}$ in three stages (Figure 1).

---

[3]SSU also relates to foundational continual learning methods that protect critical parameters, such as HAT (Serra et al., 2018), CAT (Ke et al., 2020), and SPG (Konishi et al., 2023). See Appendix E for discussions.

### 3.1 SOURCE-DRIVEN PARAMETER IMPORTANCE SCORING

The first stage of SSU scores parameter importance to identify weights critical to source model capabilities. We posit that a **source-data-driven score** is suitable, as it directly aligns with the goal of preserving source knowledge. For this purpose, we adopt the importance score from Wanda (Sun et al., 2024), a popular pruning method.[4] Using a small sample of source data $\mathcal{D}_{\text{calib}}$, Wanda computes an importance score $s_{ij}$ for each weight $\theta_{ij}$ as the product of its magnitude and the L2-norm of its corresponding input activations $X_j$: $s_{ij} = |\theta_{ij}| \cdot ||X_j||_2$. This identifies weights that are both large and consistently active. Scores are computed for all parameters in $\mathcal{M}$ except for the embeddings and language modeling head, as all these are updated during training following Hui et al. (2025).

### 3.2 COLUMN-WISE MASKING

In the second stage, SSU converts element-wise importance scores into a structured freezing mask. A structured approach is crucial because naive, element-wise freezing disrupts feature transformations and causes catastrophic forgetting (Table 3). To avoid this, SSU operates at the column level. For instance, in a forward pass $Y = WX$, freezing an entire column of the weight matrix $W$ leaves the corresponding output dimension of $Y$ unchanged, ensuring a complete feature pathway. *The approach is analogous to protecting the core structural columns of a building during renovation; the foundational support remains untouched while peripheral elements are modified.*

Mask generation begins by aggregating scores for each column. For a weight matrix $\theta \in \mathbb{R}^{d_{out} \times d_{in}}$, a column corresponds to all parameters associated with a single input feature. The total importance score $S_j$ for each column $j$ is the sum of its individual importance scores: $S_j = \sum_i s_{ij}$. $S_j$ robustly measures the contribution of each input feature, identifying the core structural columns to be preserved. For 1D parameters, such as biases, each element is treated as its own column; thus, its per-weight score $s_i$ serves as its aggregated score $S_i$.

The binary mask $\mathcal{B}$ for each weight matrix is generated by ranking columns by their $S_j$ and then selecting the top $k\%$ to freeze (50% by default following Hui et al. (2025)). The corresponding columns in the mask $\mathcal{B}$ are set to 0 (freeze), while all others are set to 1 (update).

### 3.3 CONTINUAL PRE-TRAINING

In the third stage, the model $\mathcal{M}$ is continually pre-trained on unlabeled data $\mathcal{D}_{\text{target}}$ using a standard causal language modeling objective, denoted as the loss $L$. During the backward pass, the static mask $\mathcal{B}$ is applied to the gradients, zeroing out updates for frozen columns. The gradient update rule for a weight $\theta_{ij}$ is thus $\theta_{ij} \leftarrow \theta_{ij} - \eta \cdot b_{ij} \cdot \nabla_{\theta_{ij}} L$. Here, $\eta$ is the learning rate, and $b_{ij} \in \{0, 1\}$ is the value from the mask $\mathcal{B}$ corresponding to the weight $\theta_{ij}$. This method preserves knowledge stored in the most critical input-feature pathways, thus mitigating catastrophic forgetting.

## 4 EXPERIMENTAL SETUP

### 4.1 SOURCE MODELS

Following Hui et al. (2025) who used 7B and 13B models from the same family (i.e., Llama 2), we use the 7B and 13B OLMo 2 Instruct models (Walsh et al., 2025) for our experiments. The OLMo 2 models offer strong instruction-following capabilities and fully documented training data, allowing full control and transparency in our language adaptation experiments.

### 4.2 TARGET LANGUAGES

Table 1: Source (English) and target languages. Code is based on ISO 639-1, and the language-specific ratio in Common Crawl (CC Ratio) as of CC-MAIN-2025-21.

| Language | Code | Script | Family | CC Ratio |
|---|---|---|---|---|
| English | en | Latin | Indo-European | 43.7876 |
| Nepali | ne | Devanagari | Indo-European | .0521 |
| Kyrgyz | ky | Cyrillic | Turkic | .0103 |
| Amharic | am | Ge'ez | Afro-Asiatic | .0032 |
| Hausa | ha | Latin | Afro-Asiatic | .0032 |
| Igbo | ig | Latin | Niger-Congo | .0007 |

We experiment with five typologically diverse languages (Table 1) that are significantly underrepresented in the training data of the source models but with wide availability of datasets with consistent

---

[4]While we use Wanda for its simplicity and popularity, *the SSU framework is agnostic to the importance metric.* To demonstrate this, we also evaluate two alternative source-driven scoring methods (§6).

task formulations (though data variations preclude direct performance comparisons between languages). These languages appear at least 840x less frequently than English in Common Crawl (CC),[5] which accounts for over 95% of the OLMo 2 pre-training corpus (Walsh et al., 2025).

## 4.3 CALIBRATION AND TRAINING DATA

We use tulu-3-sft-olmo-2-mixture (Lambert et al., 2025), the original instruction-tuning data for OLMo 2, for calibration (i.e., choosing which parameters to freeze). We randomly select 500 samples with a sequence length of 2,048. For CPT, we use a clean subset of MADLAD-400 (Kudugunta et al., 2023), sampling 200M tokens per language as recommended by Tejaswi et al. (2024).[6]

## 4.4 BASELINES

We compare our approach against baselines from three categories: performance benchmarks, a reference approach from a related paradigm, and state-of-the-art methods.

**Source**: Off-the-shelf OLMo 2, reporting performance without any adaptation.

**FFT**: Full fine-tuning that updates all the parameters of the model via CPT on target language data, quantifying the extent to which a model suffers from catastrophic forgetting without any intervention.

**AdaLoRA** (Zhang et al., 2023): An architecture-based method to mitigate catastrophic forgetting. This achieves the best overall performance among LoRA-like methods in Hui et al. (2025).

**HFT**: A state-of-the-art **static** selective parameter update method (Hui et al., 2025). It updates 50% of parameters using a fine-grained, per-layer strategy by randomly freezing two out of the four self-attention matrices ($W_Q, W_K, W_V, W_O$); and two out of three feed-forward matrices ($W_{up}, W_{down}, W_{gate}$) in a random half of the layers and one matrix in the remaining half. Since SSU is also a static method, HFT serves as a key baseline.

**GMT**: A state-of-the-art **dynamic** selective parameter update approach (Li et al., 2025) that drops gradients of a pre-defined ratio (50% in this study for fair comparison with HFT and SSU) with smaller absolute values on the target data.

To validate our use of source calibration data for scoring, we also introduce two calibration data-free ablation variants: (1) **SSU-Rand** that freezes an equal number of randomly-selected columns. This provides no principled way to preserve functionally important knowledge. (2) **SSU-Mag** that freezes columns based only on the magnitude score (i.e., $|\theta_{ij}|$; unlike $|\theta_{ij}| \cdot ||X_j||_2$ for SSU-Wanda), isolating the effect of the activation term.

## 4.5 EVALUATION BENCHMARKS AND METRICS

We report performance in the source and target languages across standard benchmarks.

**Chat and Instruction-following**: (1) **IFEval** (Zhou et al., 2023), reporting zero-shot accuracy (strict prompt); (2) AlpacaEval 2.0 (Li et al., 2023b; Dubois et al., 2024, **AE2**), reporting the zero-shot, length-controlled win-rate against GPT-4 (1106-preview) (OpenAI et al., 2024), with judgments from GPT-4.1 nano (2025-04-14); (3) MT-Bench (Zheng et al., 2023, **MTB**), using the mean Likert-5 score over two turns, judged by `Flow-Judge-v0.1` per the Hugging Face LightEval protocol (Fourrier et al., 2023); and (4) **GSM8K** for multi-turn, few-shot mathematical reasoning (Cobbe et al., 2021), reporting the five-shot exact match score.

**Safety**: We use the Tülu 3 safety evaluation suite (Lambert et al., 2025, **T3**) and report the macro average score in a zero-shot setting, following Lambert et al. (2025) and Walsh et al. (2025).[7]

**Source Language** (English): We evaluate target-to-English machine translation (**MT**) on FLORES-200 (NLLB Team et al., 2022), reporting three-shot chrF++ (Popović, 2017) on 500 samples, following previous work (Ahia et al., 2023; Yamaguchi et al., 2025). For summarization (**SUM**) on

---

[5]CC Ratio is based on `https://commoncrawl.github.io/cc-crawl-statistics/plots/languages`.

[6]During CPT, we remove the chat template to support unlabeled data lacking role annotations (e.g., user).

[7]As instruct models typically undergo extensive safety alignment (Gemma Team et al., 2025; Lambert et al., 2025, *inter alia.*), verifying that this is not compromised during adaptation is a crucial aspect of our analysis.

Table 2: Aggregated average performance across all languages per task. Green denotes scores better than Source with subscripts showing relative changes. **Bold** and underlined indicate best and second-best methods for each model scale. Tables 10, 11, 12, and 13 include a full suite of results.

| | Approach | Chat and Instruction-following (en) | | | | Safety | Source language (en) | | | | Target language | | | |
|---|---|---|---|---|---|---|---|---|---|---|---|---|---|---|
| | | IFEval | AE2 | MTB | GSM8K | T3 (en) | MT | SUM | MRC | MMLU | MT | SUM | MRC | MMLU |
| 7B | Source | .675$_{+0.0}$ | 32.6$_{+0.0}$ | 3.98$_{+0.0}$ | .796$_{+0.0}$ | .851$_{+0.0}$ | 30.0$_{+0.0}$ | 22.8$_{+0.0}$ | .880$_{+0.0}$ | .618$_{+0.0}$ | 20.1$_{+0.0}$ | 20.2$_{+0.0}$ | .334$_{+0.0}$ | .304$_{+0.0}$ |
| | FFT | .456$_{-32.4}$ | 10.4$_{-68.1}$ | 3.48$_{-12.5}$ | .608$_{-23.6}$ | .797$_{-6.4}$ | 42.8$_{+42.6}$ | 20.8$_{-8.7}$ | .842$_{-4.3}$ | .580$_{-6.2}$ | 30.7$_{+52.8}$ | 22.7$_{+12.4}$ | .393$_{+17.7}$ | .325$_{+6.8}$ |
| | AdaLoRA | **.669**$_{-0.8}$ | 24.6$_{-24.5}$ | 3.92$_{-1.5}$ | .721$_{-9.4}$ | .824$_{-3.2}$ | 34.1$_{+13.6}$ | 22.4$_{-1.6}$ | **.866**$_{-1.6}$ | **.602**$_{-2.6}$ | 19.9$_{-1.0}$ | 21.9$_{+8.4}$ | .318$_{-4.8}$ | .299$_{-1.8}$ |
| | HFT | .621$_{-8.0}$ | 17.6$_{-45.9}$ | 3.83$_{-3.7}$ | .677$_{-15.0}$ | .826$_{-3.0}$ | 45.2$_{+50.6}$ | 22.3$_{-2.1}$ | .854$_{-3.0}$ | .595$_{-3.7}$ | 29.8$_{+48.3}$ | 22.6$_{+11.9}$ | .377$_{+12.9}$ | .322$_{+5.8}$ |
| | GMT | .528$_{-21.7}$ | 12.5$_{-61.6}$ | 3.67$_{-7.7}$ | .635$_{-20.2}$ | .795$_{-6.6}$ | 45.5$_{+51.6}$ | 21.6$_{-5.1}$ | .841$_{-4.4}$ | .582$_{-5.8}$ | 30.9$_{+53.8}$ | 22.9$_{+13.4}$ | .385$_{+15.3}$ | .319$_{+4.8}$ |
| | SSU-Rand | .608$_{-9.9}$ | 18.0$_{-44.7}$ | 3.81$_{-4.2}$ | .683$_{-14.2}$ | .835$_{-1.9}$ | 45.5$_{+51.6}$ | 22.4$_{-1.6}$ | .861$_{-2.2}$ | .597$_{-3.4}$ | 30.2$_{+50.3}$ | 22.7$_{+12.4}$ | .394$_{+18.0}$ | .324$_{+6.4}$ |
| | SSU-Mag | .570$_{-15.5}$ | 14.9$_{-54.2}$ | 3.78$_{-5.0}$ | .655$_{-17.7}$ | .822$_{-3.4}$ | 44.7$_{+48.9}$ | 22.0$_{-3.4}$ | .859$_{-2.4}$ | .593$_{-4.1}$ | 29.7$_{+47.8}$ | 22.7$_{+12.4}$ | .383$_{+14.7}$ | .319$_{+4.8}$ |
| | SSU-Wanda | **.669**$_{-0.8}$ | **27.0**$_{-17.1}$ | **3.96**$_{-0.5}$ | **.752**$_{-5.5}$ | **.850**$_{-0.1}$ | **45.7**$_{+52.3}$ | **22.8**$_{+0.1}$ | **.869**$_{-1.3}$ | **.606**$_{-2.0}$ | **31.0**$_{+54.3}$ | **22.8**$_{+12.9}$ | **.403**$_{+20.7}$ | **.333**$_{+9.4}$ |
| 13B | Source | .763$_{+0.0}$ | 37.2$_{+0.0}$ | 4.06$_{+0.0}$ | .853$_{+0.0}$ | .821$_{+0.0}$ | 33.3$_{+0.0}$ | 24.5$_{+0.0}$ | .897$_{+0.0}$ | .665$_{+0.0}$ | 22.4$_{+0.0}$ | 20.7$_{+0.0}$ | .374$_{+0.0}$ | .329$_{+0.0}$ |
| | FFT | .448$_{-41.3}$ | 14.5$_{-61.1}$ | 3.52$_{-13.3}$ | .740$_{-13.3}$ | .737$_{-10.2}$ | 40.1$_{+20.3}$ | 15.7$_{-35.8}$ | .892$_{-0.5}$ | .647$_{-2.7}$ | 33.6$_{+50.1}$ | 22.9$_{+10.4}$ | **.492**$_{+31.6}$ | **.361**$_{+9.8}$ |
| | AdaLoRA | .719$_{-5.8}$ | 32.1$_{-13.8}$ | **4.05**$_{-0.2}$ | .815$_{-4.5}$ | .799$_{-2.7}$ | 36.6$_{+9.8}$ | **24.4**$_{-0.2}$ | **.898**$_{+0.1}$ | **.660**$_{-0.8}$ | 23.0$_{+2.7}$ | 22.3$_{+7.5}$ | .365$_{-2.4}$ | .311$_{-5.4}$ |
| | HFT | .631$_{-17.3}$ | 25.8$_{-30.7}$ | **3.92**$_{-3.4}$ | .776$_{-9.0}$ | .785$_{-4.4}$ | **44.1**$_{+32.2}$ | 20.7$_{-15.3}$ | .894$_{-0.3}$ | .658$_{-1.1}$ | .737$_{+50.5}$ | 22.8$_{+9.9}$ | .476$_{+27.3}$ | .355$_{+8.0}$ |
| | GMT | .497$_{-34.9}$ | 19.3$_{-48.2}$ | 3.64$_{-10.3}$ | .754$_{-11.6}$ | .755$_{-8.0}$ | 37.5$_{+12.5}$ | 16.5$_{-32.5}$ | .896$_{-0.1}$ | .654$_{-1.7}$ | 33.5$_{+49.6}$ | 22.8$_{+9.9}$ | .473$_{+26.5}$ | .353$_{+7.4}$ |
| | SSU-Rand | .630$_{-17.5}$ | 24.7$_{-33.7}$ | 3.89$_{-4.1}$ | .781$_{-8.5}$ | .783$_{-4.6}$ | 43.9$_{+31.6}$ | 21.7$_{-11.3}$ | **.898**$_{+0.1}$ | .656$_{-1.4}$ | 33.6$_{+50.1}$ | **23.0**$_{+10.9}$ | .478$_{+27.8}$ | .356$_{+8.3}$ |
| | SSU-Mag | .572$_{-25.1}$ | 20.6$_{-44.7}$ | 3.80$_{-6.4}$ | .763$_{-10.6}$ | .776$_{-5.5}$ | 40.2$_{+20.6}$ | 20.2$_{-17.4}$ | .892$_{-0.5}$ | .657$_{-1.2}$ | 32.8$_{+46.5}$ | 22.6$_{+8.9}$ | .467$_{+24.9}$ | .350$_{+6.5}$ |
| | SSU-Wanda | **.730**$_{-4.4}$ | **33.4**$_{-10.3}$ | **4.05**$_{-0.2}$ | **.822**$_{-3.7}$ | **.805**$_{-2.0}$ | **48.2**$_{+44.5}$ | 24.2$_{-1.0}$ | .897$_{+0.0}$ | .661$_{-0.6}$ | **34.1**$_{+52.3}$ | **23.2**$_{+11.8}$ | **.486**$_{+29.9}$ | .359$_{+9.2}$ |

XL-SUM (Hasan et al., 2021), we use zero-shot chrF++ on 500 samples. For machine reading comprehension (**MRC**) on Belebele (Bandarkar et al., 2024) and general reasoning on **MMLU** (Hendrycks et al., 2021), we report three-shot and five-shot accuracy, respectively, on their full test sets.

**Target Language**: We evaluate English-to-target **MT**, **SUM**, and **MRC** on the same target-language subsets of respective datasets and settings. For reasoning, we use Global MMLU (Singh et al., 2025) and report five-shot accuracy on its full test set.

We report average scores over three runs for generative tasks (IFEval, AE2, MTB, GSM8K, MT, SUM) and use a single deterministic run with temperature zero for classification tasks. We use language-specific prompt templates for MT, SUM, MRC, and MMLU, listed in Table 7 in the Appendix. The rest use the default prompt templates.

## 5 RESULTS

Table 2 shows performance across the four task groups: chat and instruction-following, safety, source language, and target language for all methods.

**Chat and Instruction-following.** Our SSU-Wanda achieves the best performance on all chat and instruction-following benchmarks, exhibiting the smallest average relative performance drops from Source of just 5.9% and 4.7% for the 7B and 13B models, respectively. This result is particularly important as these tasks directly measure core instruct model capabilities, such as multi-step reasoning and following complex constraints. The performance of SSU-Wanda demonstrates its efficacy in retaining source knowledge and abilities. The architecture-based method, AdaLoRA, performs second best with average degradations of 9.0% (7B) and 6.1% (13B). This aligns with previous findings that LoRA-style adaptations tend to forget less. However, as we discuss later, they also learn less from target data (Biderman et al., 2024; Hui et al., 2025).

In contrast, other methods exhibit more substantial performance drops. The state-of-the-art selective parameter update baselines lag considerably behind SSU-Wanda. For instance, the performance of HFT drops by 18.0% (7B) and 15.1% (13B), while the target-data-driven GMT degrades by 27.7% (7B) and 26.3% (13B). Notably, the static HFT method preserves source capabilities more effectively than the dynamic GMT method, supporting our main hypothesis that optimizing on signals from unstructured target data risks corrupting the foundational abilities of an instruct model (§1). The risk of standard adaptation is starkly illustrated by the overall performance of full fine-tuning (FFT). FFT suffers a drastic average performance loss of 34.1% (7B) and 32.3% (13B).

Finally, the low performance of baseline SSU variants (SSU-Rand and SSU-Mag) highlights the importance of the source-data-driven scoring. While both freezing random columns (SSU-Rand) and columns selected by magnitude alone (SSU-Mag) outperform FFT, they substantially underperform SSU-Wanda. SSU-Rand performance is 18.2% (7B) and 16.0% (13B) lower than Source, while SSU-Mag causes even greater drops of 23.0% (7B) and 21.7% (13B). The substantial underperformance of these calibration data-free approaches underscores the critical need for a source-data-informed importance scoring method for preserving the core capabilities of an instruct model in the source language. As we demonstrate in §6, this principle is not limited to Wanda; other source-data-driven scoring methods are also highly effective, confirming the versatility of the SSU framework.

**Safety.** SSU-Wanda also best preserves the safety alignment of the source, with small performance drops of only 0.1% (7B) and 2.0% (13B) compared to Source. In contrast, FFT and the target-data-driven GMT cause large drops, with safety scores dropping by up to 10.2%. While other selective methods partially preserve source performance, they still lag behind SSU-Wanda.

**Source Language.** SSU-Wanda not only preserves source language capabilities but also enhances them in the cross-lingual translation task. For the 7B model, SSU-Wanda is the top performer across all source benchmarks. For the 13B model, it ranks top in MT and MMLU and is a close second in SUM and MRC. Notably, its performance on MT (target-to-English) improves substantially by up to 52.3% relative to Source. For monolingual tasks (SUM, MRC, and MMLU), performance is almost perfectly maintained, with relative drops never exceeding 2.0% (7B) and 1.0% (13B). AdaLoRA is the second-best performer overall, also showing strong preservation across monolingual tasks. However, its gains in the MT task are substantially smaller, the worst among all approaches. This suggests that while LoRA-based methods effectively prevent forgetting, the structural isolation of their updates may be less adept at integrating new linguistic knowledge for complex cross-lingual tasks. The remaining adaptation methods generally exhibit greater performance degradation than SSU-Wanda, consistent with instruction-following and safety results.

**Target Language.** Finally, SSU-Wanda demonstrates exceptional performance on target language tasks, securing the best results across all benchmarks for both model scales in the majority of cases. Crucially, its performance is highly competitive with FFT, even surpassing it on all benchmarks for 7B models and on half for 13B models. The performance difference between SSU and FFT is consistently minimal, confirming that SSU-Wanda achieves the target-language gains of a full update with drastically smaller catastrophic forgetting. This aligns with observations from optimization theory, arguing that freezing parameters acts as a regularization term that stabilizes training and enables a sparse fine-tuned model to match or exceed the performance of its dense counterpart (Fu et al., 2023; Zhang et al., 2024b; Hui et al., 2025). All the other selective parameter update methods also yield solid improvements, though typically smaller than those of SSU-Wanda. In contrast, AdaLoRA shows the smallest improvement and often fails to surpass the source model. This confirms that LoRA-based methods have a smaller inductive bias from the target data (Biderman et al., 2024; Hui et al., 2025). This highlights the unique effectiveness of SSU-Wanda, which successfully masters tasks in the target language while preserving its original knowledge and abilities in the source.

*Overall, SSU-Wanda demonstrates the benefits of full fine-tuning without the associated catastrophic forgetting, consistently outperforming all other evaluated methods.*

## 6 ANALYSIS

This section evaluates the robustness of the SSU framework by isolating the impact of core design choices and hyperparameters. Due to resource constraints, we use the 7B model with our primary method, SSU-Wanda. We select Igbo as the target language, as it is the most underrepresented language among our target languages (Table 1).

**Parameter Freezing Ratio.** While we use a default 50% freezing ratio for fair comparison with baselines following Hui et al. (2025), this hyperparameter can impact performance. We therefore evaluate freezing ratios from 0% (defaulting to FFT) to 87.5% in 12.5% increments. As shown in Figure 2, performance on source language capabilities, such as chat and safety, generally improves as the freezing ratio increases. In contrast, performance on target language tasks often shows an

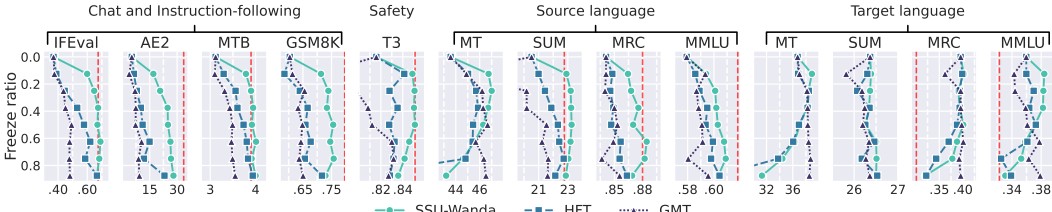

Figure 2: Model performance (SSU-Wanda, HFT, GMT) on Igbo as target language across freezing ratios. The dashed red line indicates Source performance (omitted for MT and SUM due to very low scores). Some data points for HFT and GMT are also omitted due to extremely low performance.

Table 3: Performance of different freezing strategies in SSU-Wanda and Igbo as the target language. **Bold** and underlined indicate the best and second-best methods, respectively.

| Approach | Chat and Instruction-following | | | | Safety | Source language | | | | Target language (Igbo) | | | |
|---|---|---|---|---|---|---|---|---|---|---|---|---|---|
| | IFEval | AE2 | MTB | GSM8K | T3 | MT | SUM | MRC | MMLU | MT | SUM | MRC | MMLU |
| Source | .675 | 32.6 | 3.98 | .796 | .851 | 28.5 | 22.8 | .880 | .618 | 23.0 | 23.3 | .301 | .323 |
| Column-wise (Default) | **.670** | **25.0** | **3.92** | **.756** | **.851** | 46.3 | **23.3** | **.870** | **.603** | 37.1 | 26.3 | .401 | **.371** |
| Row-wise | .548 | 11.3 | 3.74 | .675 | .846 | 46.0 | 21.8 | .862 | .598 | 36.9 | **26.5** | **.407** | .358 |
| Element-wise | .457 | 7.7 | 3.35 | .657 | .829 | **46.4** | 21.1 | .851 | .587 | **38.3** | **26.5** | .399 | .370 |

opposite trend, degrading as more parameters are frozen, with a particularly sharp drop in MMLU after reaching a 37.5% ratio. Target-to-English MT is a notable exception. Although the models generate English text, performance declines as the freezing ratio increases, particularly after 37.5%. This trend contradicts other source tasks. This occurs because MT requires knowledge of both source and target languages.

Our results show a trade-off between source knowledge retention and target language acquisition. Therefore, we recommend practitioners tailor the freezing ratio based on their specific goals: **General purpose**: A default 50% ratio offers a robust and balanced performance. **Source-capability priority**: A higher ratio (e.g., ∼ 60% or higher) is optimal, as performance on tasks like IFEval, MRC, and MMLU plateaus around this point. **Target-language priority**: A lower ratio (e.g., ∼ 40% or lower) is preferable, given the performance drops observed in MT and MMLU beyond this threshold.

**Impact of Freezing Ratio on Baselines.** We extend this analysis to state-of-the-art selective parameter update baselines (Figure 2). The closest baseline, the static method HFT, follows a trend similar to SSU but fails to surpass the performance of SSU across tasks and freezing ratios. In contrast, the dynamic method GMT exhibits a different trend. While it often achieves strong target language and MT performance at ratios above 60%, it consistently yields low performance on monolingual source tasks regardless of the freezing ratio. We attribute this to the dynamic nature of GMT, which allows updates to any parameter over time, leading to cumulative corruption from unstructured target data optimization (§1 and §5). Ultimately, this confirms SSU as the optimal method for simultaneously achieving strong source preservation and high target language gains.

**Alternative Freezing Methods.** SSU employs column-wise freezing to preserve the entire processing pathway of critical source features (§3.2). To validate this design choice, we compare its effectiveness against row-wise and element-wise freezing. As shown in Table 3, the results demonstrate a clear advantage for our column-wise approach. Column-wise freezing consistently achieves the best performance on chat, safety, and source language tasks.[8] On target language tasks, it remains highly competitive, with only a 1.2 point drop on MT compared to element-wise freezing. These results validate the guiding hypothesis for the design of SSU: *preserving entire feature pathways is a critical strategy to safeguard source knowledge while enabling effective target-language adaptation.*

---

[8]While row-wise freezing preserves all connections from a single input neuron, it fails to protect any single, complete output feature. This explains its weaker performance across chat, safety, and source language tasks.

Table 4: Performance of different SSU importance scoring methods using Igbo as the target. **Bold** and underlined denote best and second-best adaptation approaches with relative changes in subscripts.

| Approach | Chat and Instruction-following | | | | Safety | Source language | | | | Target language (Igbo) | | | |
|---|---|---|---|---|---|---|---|---|---|---|---|---|---|
| | IFEval | AE2 | MTB | GSM8K | T3 | MT | SUM | MRC | MMLU | MT | SUM | MRC | MMLU |
| Source | .675 $_{+0.0}$ | 32.6 $_{+0.0}$ | 3.98 $_{+0.0}$ | .796 $_{+0.0}$ | .851 $_{+0.0}$ | 28.5 $_{+0.0}$ | 22.8 $_{+0.0}$ | .880 $_{+0.0}$ | .618 $_{+0.0}$ | 23.0 $_{+0.0}$ | 23.3 $_{+0.0}$ | .301 $_{+0.0}$ | .323 $_{+0.0}$ |
| SSU-Rand | .564 $_{-16.4}$ | 12.5 $_{-61.6}$ | 3.75 $_{-5.7}$ | .680 $_{-14.6}$ | .838 $_{-1.6}$ | 45.9 $_{+61.3}$ | 22.4 $_{-1.6}$ | .856 $_{-2.7}$ | .597 $_{-3.4}$ | **37.3** $_{+62.5}$ | 26.4 $_{+13.4}$ | **.401** $_{+33.2}$ | .355 $_{+10.0}$ |
| SSU-Mag | .497 $_{-26.3}$ | 8.9 $_{-72.7}$ | 3.59 $_{-9.8}$ | .638 $_{-19.9}$ | .828 $_{-2.7}$ | 45.1 $_{+58.5}$ | 21.7 $_{-4.7}$ | .852 $_{-3.2}$ | .592 $_{-4.2}$ | 36.6 $_{+59.5}$ | 26.2 $_{+12.5}$ | .379 $_{+25.9}$ | .348 $_{+7.8}$ |
| SSU-Wanda (Default) | .670 $_{-0.7}$ | 25.0 $_{-23.2}$ | 3.92 $_{-1.5}$ | **.756** $_{-5.0}$ | **.851** $_{-0.0}$ | 46.3 $_{+62.7}$ | **23.3** $_{+2.3}$ | .870 $_{-1.1}$ | .603 $_{-2.4}$ | 37.1 $_{+61.7}$ | 26.3 $_{+12.9}$ | **.401** $_{+33.2}$ | .371 $_{+14.9}$ |
| SSU-SparseGPT | **.678** $_{+0.5}$ | 24.5 $_{-24.8}$ | 3.89 $_{-2.2}$ | .751 $_{-5.7}$ | .843 $_{-1.0}$ | 46.2 $_{+62.3}$ | 23.1 $_{+1.4}$ | **.876** $_{-0.5}$ | .604 $_{-2.3}$ | 37.2 $_{+62.1}$ | **26.5** $_{+13.8}$ | .400 $_{+32.8}$ | **.372** $_{+15.2}$ |
| SSU-FIM | .669 $_{-0.8}$ | **26.3** $_{-19.2}$ | **3.94** $_{-1.0}$ | .747 $_{-6.2}$ | .847 $_{-0.5}$ | **46.4** $_{+63.0}$ | 23.2 $_{+1.9}$ | .874 $_{-0.7}$ | **.609** $_{-1.5}$ | 37.1 $_{+61.7}$ | **26.5** $_{+13.8}$ | .399 $_{+32.5}$ | .371 $_{+14.9}$ |

Table 5: Performance of SSU-Wanda with different calibration data sources and sizes, using Igbo as the target language.

| Approach | Chat and Instruction-following | | | | Safety | Source language | | | | Target language (Igbo) | | | |
|---|---|---|---|---|---|---|---|---|---|---|---|---|---|
| | IFEval | AE2 | MTB | GSM8K | T3 | MT | SUM | MRC | MMLU | MT | SUM | MRC | MMLU |
| Source | .675 | 32.6 | 3.98 | .796 | .851 | 28.5 | 22.8 | .880 | .618 | 23.0 | 23.3 | .301 | .323 |
| Default (500 examples) | .670 | 25.0 | 3.92 | .756 | .851 | 46.3 | 23.3 | .870 | .603 | 37.1 | 26.3 | .401 | .371 |
| Alpaca (500 examples) | .673 | 24.0 | 3.97 | .750 | .849 | 46.7 | 23.1 | .874 | .604 | 37.1 | 26.2 | .394 | .379 |
| Default (128 examples) | .682 | 24.3 | 3.89 | .754 | .852 | 46.4 | 23.2 | .873 | .600 | 37.2 | 26.3 | .410 | .371 |

**Alternative Importance Scoring Methods.** SSU is compatible with alternative importance scoring methods beyond Wanda. To demonstrate this, we evaluate two different source-data-driven methods: SparseGPT (Frantar & Alistarh, 2023) and the diagonal of the Fisher Information Matrix (Kirkpatrick et al., 2017, FIM); see Appendix B for details. In monolingual source tasks, SSU-SparseGPT and SSU-FIM show comparable average performance drops (4.3% and 3.5%, respectively) to SSU-Wanda (4.0%), as detailed in Table 4. This contrasts sharply with the larger drops of data-free variants like SSU-Rand (13.5%) and SSU-Mag (17.9%). These findings demonstrate the versatility of SSU, offering strong performance across various source-data-driven scoring methods.

**Calibration Data for Parameter Importance Scoring.** SSU-Wanda requires source calibration data to identify critical model weights since it relies on Wanda for parameter importance scoring. While we use the original instruction-tuning data for OLMo 2 in our main experiments, this is often unavailable for other frontier models. We therefore investigate the efficacy of using an alternative, publicly available dataset. Specifically, we use Alpaca (Taori et al., 2023) as the calibration dataset and follow the exact same preprocessing and training procedures as the original data. Table 5 shows that performance with Alpaca is highly comparable to that with the original data, with a maximum difference of only 1.0, demonstrating the robustness of SSU-Wanda to the choice of calibration data.

**Calibration Data Size for Parameter Importance Scoring.** SSU uses 500 source calibration examples by default to compute parameter importance scores (§4.3). To assess sensitivity to this hyperparameter, we compare the default (500 examples, ∼1M tokens) with a smaller 128-example set (∼0.26M tokens), a size common in model pruning literature (Williams & Aletras, 2024). The results in Table 5 show minimal changes across tasks; the maximum performance difference observed is only 1.2 points on IFEval. This confirms the robustness of SSU to calibration data size, demonstrating that a small sample set suffices for effective importance scoring.

**Comparison to Additional Baselines.** We also compare SSU-Wanda to other selective parameter update methods: LoTA (Panda et al., 2024) and S2FT (Yang et al., 2024), using their default configurations. We evaluate LoTA at its default 90% sparsity and at 50% sparsity to match the freezing ratio of SSU. For S2FT, we test its default down projection-focused adaptation. As shown in Table 6, LoTA at 90% sparsity exhibits inferior source preservation compared to SSU-Wanda (e.g., 7.6% vs. 4.0% average drop on monolingual source tasks) and lower target gains (23.9% vs. 30.7%). While LoTA at 50% sparsity achieves substantial target gains (31.7%), it suffers severe catastrophic forgetting on monolingual source tasks (19.9% drop). S2FT effectively preserves source capabilities (3.3% drop) but yields minimal target gains (2.3%). These results underscore that only SSU-Wanda

Table 6: Performance of additional adaptation baselines: LoTA and S2FT using Igbo as the target. **Bold** and underlined denote best and second-best adaptation approaches with relative changes in subscripts. More results are in Appendix D.

| Approach | Chat and Instruction-following | | | | Safety | Source language | | | | Target language (Igbo) | | | |
|---|---|---|---|---|---|---|---|---|---|---|---|---|---|
| | IFEval | AE2 | MTB | GSM8K | T3 | MT | SUM | MRC | MMLU | MT | SUM | MRC | MMLU |
| Source | $.675_{+0.0}$ | $32.6_{+0.0}$ | $3.98_{+0.0}$ | $.796_{+0.0}$ | $.851_{+0.0}$ | $28.5_{+0.0}$ | $22.8_{+0.0}$ | $.880_{+0.0}$ | $.618_{+0.0}$ | $23.0_{+0.0}$ | $23.3_{+0.0}$ | $.301_{+0.0}$ | $.323_{+0.0}$ |
| SSU-Wanda | $\underline{.670}_{-0.7}$ | $\underline{25.0}_{-23.2}$ | $3.92_{-1.5}$ | $\underline{.756}_{-5.0}$ | $\mathbf{.851}_{-0.0}$ | $\mathbf{46.3}_{+62.7}$ | $\mathbf{23.3}_{+2.3}$ | $\underline{.870}_{-1.1}$ | $.603_{-2.4}$ | $\underline{37.1}_{+61.7}$ | $\underline{26.3}_{+12.9}$ | $\underline{.401}_{+33.2}$ | $\underline{.371}_{+14.9}$ |
| LoTA (90% Sparsity) | $.638_{-5.4}$ | $20.4_{-37.4}$ | $\mathbf{3.98}_{+0.0}$ | $.706_{-11.3}$ | $.827_{-2.8}$ | $45.2_{+58.8}$ | $\underline{22.7}_{-0.3}$ | $.864_{-1.8}$ | $\mathbf{.606}_{-2.0}$ | $34.4_{+49.9}$ | $26.2_{+12.5}$ | $.366_{+21.5}$ | $.360_{+11.5}$ |
| LoTA (50% Sparsity) | $.449_{-33.4}$ | $8.3_{-74.5}$ | $3.45_{-13.3}$ | $.636_{-20.1}$ | $.824_{-3.2}$ | $\underline{45.8}_{+60.9}$ | $21.5_{-5.6}$ | $.844_{-4.1}$ | $.590_{-4.6}$ | $\mathbf{37.8}_{+64.7}$ | $\mathbf{26.4}_{+13.4}$ | $\mathbf{.402}_{+33.5}$ | $\mathbf{.372}_{+15.2}$ |
| S2FT (Down) | $\mathbf{.695}_{+3.0}$ | $\mathbf{27.9}_{-14.3}$ | $\underline{3.99}_{+0.3}$ | $\mathbf{.732}_{-8.0}$ | $\underline{.834}_{-2.0}$ | $36.7_{+29.0}$ | $22.6_{-0.7}$ | $\mathbf{.857}_{-2.6}$ | $\underline{.603}_{-2.4}$ | $21.7_{-5.4}$ | $26.0_{+11.6}$ | $.303_{+0.6}$ | $.331_{+2.5}$ |

achieves both strong source preservation and high target language gains comparable to FFT. This trend holds across different hyperparameter settings (see Appendix D).

**Qualitative Analysis.** SSU-Wanda surpasses other state-of-the-art selective parameter update baselines across all chat and instruction-following benchmarks (Table 2). This performance gap arises partly because baseline methods are prone to code-mixing (i.e., the unintentional blending of multiple languages in responses) or generating responses entirely in the target language, despite English instructions. A typical example of this behavior for models trained on Igbo is as below:

Analyzing the language ratio in generated responses on AE2 shows that SSU produces code-mixed text in only 1.0% of its responses on average for the 7B models. In contrast, HFT and GMT generate code-mixed text in 6.4% and 16.9%, respectively.[9] This substantial reduction in the occurrence of code-mixing reflects the more robust retention of the source language abilities and superior chat and instruction-following performance of SSU.

> Instruction in EN: How do I take care of a wooden table?
> HFT Response: *To take care nke a wood table, clean ya na a soft duster ma ọ bụ microfiber towel iji wepụ dust na grime. N'ihe banyere stains, mix otu teaspoon nke mild detergent na mmiri na-a teaspoon n'ime mmiri...* (continued.)
> SSU Response: *To take care, clean your wooden table regularly with mild soap and water. Use a soft cloth for polishing, applying wood polish or beeswax to maintain its natural finish. Avoid placing hot items directly on the surface to prevent scratches. Keep it away from direct sunlight and excessive moisture.*

**Theoretical Analysis.** SSU addresses the stability-plasticity dilemma in neural systems (Grossberg, 1982), balancing plasticity for new knowledge with stability for prior knowledge. SSU mitigates catastrophic forgetting by identifying and freezing a source-critical subnetwork, extending the Lottery Ticket Hypothesis (Frankle & Carbin, 2019) to transfer learning. Using an importance score to shield crucial parameters, SSU proactively preserves source capabilities by enforcing a hard constraint that confines updates to a subspace that avoids interfering with the source language knowledge. This approach aligns with findings on spurious forgetting (Zheng et al., 2025), which suggest that performance drops can stem from task misalignment caused by nearly orthogonal weight updates. Furthermore, SSU employs structured, column-wise masking, motivated by the need to preserve entire learned features. Unlike unstructured pruning which can degrade learned representations, pruning entire columns of a weight matrix corresponds to removing specific neurons or feature detectors (Voita et al., 2019). This structural preservation ensures that the core feature space of the source model remains intact, enabling effective adaptation to the target language.

# 7 CONCLUSION

We introduced **S**ource-**S**hielded **U**pdates (**SSU**) for language adaptation of instruct models using only unlabeled target language data. SSU is a framework that proactively identifies critical source knowledge using an importance scoring method and a small set of source calibration data. It then shields this knowledge via a column-wise freezing strategy before adaptation, effectively preventing catastrophic forgetting in the source language. Extensive experiments across five languages and two model scales show that SSU best preserves crucial source capabilities, such as instruction-following and safety, over strong baselines while achieving target language proficiency matching or surpassing full fine-tuning. This work provides an effective and scalable pathway to expand the linguistic reach of instruct models without costly, specialized data, opening avenues for robust model adaptation.

---

[9]We use GlotLID (Kargaran et al., 2023, Commit 28d4264) to compute the language ratio of each response. If the normalized confidence for English is less than 0.9, it is regarded as code-mixed.

ETHICS STATEMENT

The authors acknowledge the use of Large Language Models (LLMs) during the preparation of this work. Gemini 2.5 Pro was utilized to find related work and to improve the grammar and clarity of the draft. Additionally, GPT-5 served as a coding assistant for implementation and debugging.

REPRODUCIBILITY STATEMENT

Our code and a step-by-step guide for preprocessing, training, evaluation, and analysis for both the proposed method and all baselines are available on an anonymous GitHub repository: https://anonymous.4open.science/r/ssu-iclr-2026/. The repository reflects updates made on November 20, 2025, to include the additional baselines: LoTA and S2FT. This resource will remain accessible until the ICLR 2026 decision notification date: January 22, 2026 (AOE). Full details on hyperparameters, software, and hardware, including specific versions used, are provided in Appendix B.

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

---

**Appendix Directory**

- **Appendix A:** Evaluation Details
- **Appendix B:** Implementation Details
    - General Setup
    - Alternative Scoring Method Implementations
- **Appendix C:** Supplementary Results
- **Appendix D:** Supplementary Analysis
- **Appendix E:** Extended Related Work

---

## A   EVALUATION DETAILS

Table 7 shows language-specific prompt templates for each task.

## B   IMPLEMENTATION DETAILS

### B.1   GENERAL SETUP

**Hyperparameters.**   Tables 8 and 9 list the hyperparameters in CPT and evaluation, respectively.

**Software.**   We use HF datasets (Lhoest et al., 2021, v3.6.0) for preprocessing, HF transformers (Wolf et al., 2020, v4.52.4), HF peft (Mangrulkar et al., 2022, v0.15.2), FlashAttention-2 (Dao, 2024, v2.7.4) and PyTorch (Ansel et al., 2024, v2.6.0) for training. We use lm-evaluation-harness (Gao et al., 2023, v0.4.8) for IFEval and GSM8K evaluation, alpaca-eval (Li et al., 2023b, v0.6.6) for AE2 evaluation, Ai2 Safety Tool for T3 evaluation,[10] and HF LightEval (Fourrier et al., 2023, Commit 327071f) for the rest.

**Hardware.**   We mainly use a single AMD MI300X GPU with ROCm 6.4.1 for experiments. Additionally, we use either a single NVIDIA H100 80GB, A100 80GB, or A100 40GB GPU with CUDA 12.9 for evaluation.

### B.2   ALTERNATIVE SCORING METHOD IMPLEMENTATIONS

**SSU-SparseGPT.**   This method employs a metric from Frantar & Alistarh (2023) that approximates second-order information. The score for any weight $\theta_{ij}$ in an input column $j$ is the average squared activation of the corresponding input neuron: $s_{ij} = \mathbb{E}_{x \in \mathcal{D}_{\text{calib}}} x_j^2$.

**SSU-FIM.**   This method uses the diagonal of the Fisher Information Matrix, which measures output sensitivity to parameter changes (Kirkpatrick et al., 2017). We approximate the Fisher score for a parameter $\theta_{ij}$ as the average squared gradient of the negative log-likelihood loss $L$ over $\mathcal{D}_{\text{calib}}$: $s_{ij} = \mathbb{E}_{(x,y) \in \mathcal{D}_{\text{calib}}} (\frac{\partial L}{\partial \theta_{ij}})^2$.

## C   SUPPLEMENTARY RESULTS

Tables 10, 11, 12, and 13 show performances on English chat and instruction-following benchmarks, English safety alignment benchmark, general English benchmarks, and general target language benchmarks, respectively. Results for IFEval, AE2, MTB, GSM9K, MT, and SUM are averaged across three different runs. The rest are single-run results as they are evaluated in a deterministic-manner.

---

[10]Following Lambert et al. (2025), we use their forked version: `https://github.com/nouhadziri/safety-eval-fork` (Commit 2920bb8).

Table 7: Language-specific prompt templates. We generate the templates for each target language using a machine translation API, following Yong et al. (2023).

| Task | Language | Template |
|---|---|---|
| X-En MT | English | Translate {X: a target language} to English: {sentence} = |
| | Nepali | नेपालीलाई अङ्ग्रेजीमा अनुवाद गर्नुहोस्: {sentence} = |
| | Kyrgyz | Кыргызчадан англисчеге которуу: {sentence} = |
| | Amharic | አማርኛን ወደ እንግሊዝኛ ተርጉም: {sentence} = |
| | Hausa | Fassara Hausa zuwa Turanci: {sentence} = |
| | Igbo | Sụgharịa Igbo gaa na Bekee: {sentence} = |
| En-X MT | English | Translate English to X: {sentence} = |
| | Nepali | अङ्ग्रेजीलाई नेपालीमा अनुवाद गर्नुहोस्: {sentence} = |
| | Kyrgyz | Англисчеден кыргызчага которуу: {sentence} = |
| | Amharic | እንግሊዝኛን ወደ አማርኛ ተርጉም: {sentence} = |
| | Hausa | Fassara Turanci zuwa Hausa: {sentence} = |
| | Igbo | Sụgharịa Bekee gaa n'Igbo: {sentence} = |
| SUM | English | Summarize the following text in English: {text} Summary: |
| | Nepali | तलको पाठलाई नेपालीमा संक्षेपमा लेख्नुहोस्: {text} सारांश: |
| | Kyrgyz | Төмөнкү текстти кыргызча кыскача жазыңыз: {text} Кыскача: |
| | Amharic | የታችኛው ጽሁፍን በአማርኛ አጭር በማድረግ አሳትረኸ: {text} አጭር ማግለጫ: |
| | Hausa | Takaita rubutu mai zuwa cikin Hausa: {text} Takaitawa: |
| | Igbo | Chịkọta edemede a n'Igbo: {text} Nchịkọta: |
| MRC | English | {context} Question: {question} A. {option A} B. {option B} C. {option C} D. {option D} Answer: |
| | Nepali | {context} प्रश्न: {question} A. {option A} B. {option B} C. {option C} D. {option D} उत्तर: |
| | Kyrgyz | {context} Суроо: {question} A. {option A} B. {option B} C. {option C} D. {option D} Жооп: |
| | Amharic | {context} ጥያቄ: {question} A. {option A} B. {option B} C. {option C} D. {option D} መልስ: |
| | Hausa | {context} Tambaya: {question} A. {option A} B. {option B} C. {option C} D. {option D} Amsa: |
| | Igbo | {context} Ajụjụ: {question} A. {option A} B. {option B} C. {option C} D. {option D} Aziza: |
| MMLU | English | The following are multiple choice questions (with answers) about {subject}. {context} Question: {question} A. {option A} B. {option B} C. {option C} D. {option D} Answer: |
| | Nepali | तल {subject} सम्बन्धी बहु-विकल्प प्रश्नहरू (उत्तर सहित) दिइएका छन्। {context} प्रश्न: {question} A. {option A} B. {option B} C. {option C} D. {option D} उत्तर: |
| | Kyrgyz | Бул {subject} боюнча бир нече тандоо суроолору (жооптор менен) төмөндө келтирилген. {context} Суроо: {question} A. {option A} B. {option B} C. {option C} D. {option D} Жооп: |
| | Amharic | ከታች ስለ {subject} የቀረቡ ባለብዙ ምርጫ ጥያቄዎች (ከመልሶች ጋር) ናቸው። {context} ጥያቄ: {question} A. {option A} B. {option B} C. {option C} D. {option D} መልስ: |
| | Hausa | Wadannan tambayoyi masu zabi da yawa (tare da amsoshi) game da {subject} ne. {context} Tambaya: {question} A. {option A} B. {option B} C. {option C} D. {option D} Amsa: |
| | Igbo | Nke a bụ ajụjụ ọnụ nhọrọ ọtụtụ (na aziza) gbasara {subject}. {context} Ajụjụ: {question} A. {option A} B. {option B} C. {option C} D. {option D} Aziza: |

Table 8: Hyperparameters for continual pre-training. Values for GMT and AdaLoRA were selected based on our setup, as they were not provided in their respective original papers (Li et al., 2025; Hui et al., 2025).

| Hyperparameters | Values |
|---|---|
| Batch size | 32 |
| Number of training steps | 12,208 |
| Optimizer | adamw_apex_fused |
| Adam $\epsilon$ | 1e-8 |
| Adam $\beta_1$ | 0.9 |
| Adam $\beta_2$ | 0.999 |
| Sequence length | 512 |
| Learning rate | 5e-5 |
| Learning rate scheduler | cosine |
| Warmup steps | First 5% of steps |
| Weight decay | 0.01 |
| Attention dropout | 0.0 |
| Training precision | BF16 |
| **HFT, GMT, SSU** | |
| Target freezing ratio | 0.5 |
| **GMT** | |
| Accumulation interval | 4 |
| **AdaLoRA** | |
| Target $r$ | 8 |
| LoRA $\alpha$ | 32 |
| LoRA dropout | 0.05 |
| $T_{\text{init}}$ | 1,000 |
| $T_{\text{final}}$ | 8,546 |
| $\delta_t$ | 20 |
| LoRA $\beta_1$ | 0.85 |
| LoRA $\beta_2$ | 0.85 |
| Coefficient of orthogonal regularization | 0.5 |
| **LoTA** | |
| Mask calibration steps | 100 |
| **S2FT** | |
| $d_{\text{ratio}}$ (Down) | 0.015 (equivalent to LoRA $r = 8$) |
| $o_{\text{ratio}}$ (Output) | 0.015 (equivalent to LoRA $r = 8$) |

Table 9: Parameters for generation tasks. N/A for GSM8K indicates that a model generates text until it detects default stop symbols or reaches its maximum sequence length.

| Parameters | Values |
|---|---|
| Temperature | 0.8 |
| Repetition penalty | 1.1 |
| Top $k$ | 40 |
| Top $p$ | 0.9 (MT, SUM, MTBench) |
| | 0.8 (AE2, IFEval, GSM8K) |
| Sampling | True |
| Max. generated tokens | 128 (MT, SUM) |
| | 512 (AE2) |
| | 1,024 (MTBench) |
| | 1,280 (IFEval) |
| | N/A (GSM8K) |

Table 10: Performance on chat and instruction-following tasks in English. The best and second-best adaptation approaches for each model scale are indicated in **bold** and underlined, respectively.

| | Approach | IFEval | | | | | AE2 | | | | | MTB | | | | | GSM8K | | | | |
|---|---|---|---|---|---|---|---|---|---|---|---|---|---|---|---|---|---|---|---|---|---|
| | | ne | ky | am | ha | ig | ne | ky | am | ha | ig | ne | ky | am | ha | ig | ne | ky | am | ha | ig |
| | Source | .675 | .675 | .675 | .675 | .675 | 32.6 | 32.6 | 32.6 | 32.6 | 32.6 | 3.98 | 3.98 | 3.98 | 3.98 | 3.98 | .796 | .796 | .796 | .796 | .796 |
| 7B | FFT | .520 | .480 | .495 | .417 | .369 | 14.3 | 12.6 | 12.1 | 7.8 | 5.2 | 3.80 | 3.50 | 3.60 | 3.40 | 3.12 | .623 | .619 | .593 | .602 | .604 |
| | AdaLoRA | **.668** | **.679** | **.681** | .646 | .669 | 27.2 | 25.7 | 25.7 | **24.6** | 20.0 | **3.98** | 3.96 | 3.89 | **3.92** | 3.87 | .736 | .742 | .737 | .704 | .685 |
| | HFT | .636 | .652 | .636 | .604 | .578 | 22.6 | 18.3 | 21.0 | 15.1 | 11.1 | 3.95 | 3.82 | 3.85 | 3.77 | 3.73 | .699 | .689 | .692 | .646 | .659 |
| | GMT | .596 | .571 | .577 | .405 | .492 | 17.7 | 14.2 | 16.1 | 7.3 | 7.3 | 3.92 | 3.74 | 3.79 | 3.44 | 3.49 | .671 | .607 | .645 | .606 | .648 |
| | SSU-Rand | .619 | .624 | .634 | .599 | .564 | 24.0 | 19.1 | 19.8 | 14.8 | 12.5 | 3.86 | 3.81 | 3.87 | 3.79 | 3.75 | .701 | .678 | .693 | .660 | .680 |
| | SSU-Mag | .595 | .617 | .591 | .548 | .497 | 19.2 | 16.8 | 18.3 | 11.5 | 8.9 | 3.87 | 3.86 | 3.81 | 3.79 | 3.59 | .682 | .665 | .660 | .629 | .638 |
| | SSU-Wanda | .655 | .664 | .661 | **.688** | **.670** | 28.1 | 28.7 | 28.5 | 24.6 | 25.0 | **4.02** | **4.02** | **3.96** | 3.91 | **3.92** | **.746** | **.759** | **.749** | **.741** | **.756** |
| | Source | .763 | .763 | .763 | .763 | .763 | 37.2 | 37.2 | 37.2 | 37.2 | 37.2 | 4.06 | 4.06 | 4.06 | 4.06 | 4.06 | .853 | .853 | .853 | .853 | .853 |
| 13B | FFT | .549 | .468 | .506 | .405 | .314 | 23.6 | 14.7 | 18.6 | 11.9 | 3.7 | 3.91 | 3.66 | 3.69 | 3.43 | 2.93 | .768 | .730 | .732 | .733 | .737 |
| | AdaLoRA | **.720** | **.733** | **.737** | .728 | .675 | 34.6 | **34.1** | **33.2** | 30.0 | 28.7 | **4.10** | 4.08 | **4.09** | 4.03 | 3.94 | .812 | .814 | .812 | **.821** | .815 |
| | HFT | .693 | .680 | .676 | .578 | .528 | 31.2 | 29.1 | 27.4 | 23.4 | 17.9 | 4.08 | 4.04 | 3.99 | 3.84 | 3.69 | .802 | .793 | .762 | .760 | .765 |
| | GMT | .628 | .527 | .543 | .404 | .381 | 28.1 | 20.1 | 19.8 | 16.2 | 12.3 | 3.91 | 3.89 | 3.54 | 3.55 | 3.34 | .787 | .759 | .688 | .763 | .771 |
| | SSU-Rand | .672 | .703 | .677 | .558 | .539 | 30.2 | 28.2 | 26.8 | 21.9 | 16.2 | 3.97 | 3.97 | 3.98 | 3.85 | 3.66 | .787 | .795 | .777 | .766 | .780 |
| | SSU-Mag | .651 | .648 | .636 | .489 | .434 | 28.3 | 24.8 | 23.5 | 16.8 | 9.7 | 4.00 | 3.93 | 3.98 | 3.76 | 3.35 | .782 | .768 | .755 | .756 | .751 |
| | SSU-Wanda | .718 | .723 | .733 | **.739** | **.739** | **34.7** | 33.7 | 32.2 | **33.8** | **32.8** | 4.04 | **4.11** | 4.01 | **4.10** | **4.01** | **.831** | **.827** | **.814** | .808 | **.830** |

Table 11: Performance on Tülu 3 safety evaluation suite (T3). The best and second-best adaptation approaches for each model scale are indicated in **bold** and underlined, respectively.

| | Approach | T3 (↑) | | | | |
|---|---|---|---|---|---|---|
| | | ne | ky | am | ha | ig |
| | Source | .851 | .851 | .851 | .851 | .851 |
| 7B | FFT | .770 | .791 | .800 | .807 | .816 |
| | AdaLoRA | **.842** | .829 | .836 | .806 | .805 |
| | HFT | .812 | .816 | .839 | .833 | .828 |
| | GMT | .777 | .791 | .811 | .782 | .812 |
| | SSU-Rand | .824 | .838 | .841 | .832 | .838 |
| | SSU-Mag | .811 | .813 | .831 | .829 | .828 |
| | SSU-Wanda | **.842** | **.846** | **.855** | **.856** | **.851** |
| | Source | .821 | .821 | .821 | .821 | .821 |
| 13B | FFT | .745 | .710 | .792 | .657 | .782 |
| | AdaLoRA | **.816** | **.805** | .815 | .759 | .799 |
| | HFT | .790 | .743 | .817 | .764 | .812 |
| | GMT | .756 | .735 | .751 | .736 | .798 |
| | SSU-Rand | .798 | .756 | .792 | .768 | .799 |
| | SSU-Mag | .774 | .742 | .804 | .747 | .811 |
| | SSU-Wanda | .809 | .789 | **.819** | **.797** | **.813** |

Table 12: Performance on source language (English) tasks. Scores that are better than Source are highlighted in green . The best and second-best adaptation approaches for each model scale are indicated in **bold** and underlined, respectively.

| | Approach | MT | | | | | SUM | | | | | MRC | | | | | MMLU | | | | |
|---|---|---|---|---|---|---|---|---|---|---|---|---|---|---|---|---|---|---|---|---|---|
| | | ne | ky | am | ha | ig | ne | ky | am | ha | ig | ne | ky | am | ha | ig | ne | ky | am | ha | ig |
| | Source | 45.4 | 28.8 | 19.5 | 27.9 | 28.5 | 22.8 | 22.8 | 22.8 | 22.8 | 22.8 | .880 | .880 | .880 | .880 | .880 | .618 | .618 | .618 | .618 | .618 |
| 7B | FFT | 49.5 | **44.2** | 28.0 | 48.6 | 43.6 | 21.8 | 20.6 | 20.1 | 21.1 | 20.5 | .842 | .829 | .852 | .843 | .841 | .574 | .582 | .586 | .578 | .579 |
| | AdaLoRA | 47.6 | 33.1 | 14.1 | 39.8 | 36.2 | 22.4 | 22.9 | **22.6** | 22.1 | 22.1 | **.874** | **.878** | .871 | .860 | .847 | **.608** | **.614** | **.611** | .585 | .593 |
| | HFT | 52.5 | 43.7 | 35.8 | 48.4 | 45.4 | 22.6 | 22.7 | 22.0 | 22.1 | 22.3 | .858 | .863 | .857 | .846 | .847 | .596 | .597 | .604 | .586 | .594 |
| | GMT | 50.3 | 43.7 | 37.8 | 49.1 | 46.7 | 22.4 | 22.2 | 21.6 | 20.5 | 21.5 | .850 | .818 | .856 | .829 | .853 | .579 | .578 | .599 | .565 | .591 |
| | SSU-Rand | 51.6 | 44.1 | 36.4 | 49.4 | 45.9 | 22.7 | 22.8 | 22.1 | 22.2 | 22.4 | .858 | .864 | .872 | .856 | .856 | .600 | .599 | .605 | .584 | .597 |
| | SSU-Mag | 51.4 | 43.4 | 35.8 | 47.9 | 45.1 | 22.5 | 22.0 | 21.9 | 22.1 | 21.7 | .863 | .864 | .867 | .849 | .852 | .592 | .595 | .607 | .581 | .592 |
| | SSU-Wanda | 52.3 | 43.9 | 36.4 | 49.7 | 46.3 | 22.7 | 23.1 | 22.2 | 22.9 | 23.3 | .871 | .868 | .874 | .863 | .870 | .606 | .608 | .609 | .605 | .603 |
| | Source | 50.7 | 30.5 | 22.7 | 31.0 | 31.9 | 24.5 | 24.5 | 24.5 | 24.5 | 24.5 | .897 | .897 | .897 | .897 | .897 | .665 | .665 | .665 | .665 | .665 |
| 13B | FFT | 49.7 | 39.2 | 39.2 | 43.5 | 28.8 | 21.5 | 8.6 | 19.0 | 14.4 | 14.8 | .890 | .891 | **.901** | .891 | .889 | .650 | .643 | .657 | .650 | .637 |
| | AdaLoRA | 52.1 | 33.1 | 19.8 | 40.6 | 37.2 | 24.1 | **25.6** | **24.4** | **24.7** | 23.4 | **.906** | **.901** | .898 | .894 | **.892** | **.662** | **.663** | .662 | **.660** | .651 |
| | HFT | 55.1 | 38.6 | 41.6 | 50.1 | 35.1 | 24.5 | 20.5 | 22.7 | 16.8 | 18.8 | .897 | .896 | .893 | **.899** | .888 | .659 | .652 | **.665** | .657 | .655 |
| | GMT | 48.7 | 37.1 | 23.2 | 45.2 | 33.4 | 23.4 | 12.9 | 15.9 | 14.1 | 16.4 | .892 | .893 | .900 | .896 | **.897** | .653 | .658 | .660 | .654 | .643 |
| | SSU-Rand | 54.4 | 39.7 | 36.3 | 49.7 | 39.6 | 24.9 | 23.6 | 22.9 | 16.6 | 20.4 | .897 | **.903** | .900 | .897 | .891 | .658 | .654 | .663 | .653 | .653 |
| | SSU-Mag | 53.4 | 37.4 | 32.5 | 45.9 | 31.5 | 24.4 | 20.6 | 20.7 | 16.8 | 18.6 | .893 | .896 | .896 | .894 | .883 | .659 | .656 | .662 | .659 | .647 |
| | SSU-Wanda | 55.7 | 45.1 | 43.8 | 51.4 | 45.1 | 24.4 | 25.3 | 24.0 | 23.8 | 23.8 | .898 | .901 | .893 | .898 | .897 | .662 | .660 | .664 | .659 | .659 |

Table 13: Performance on target language tasks. Scores that are better than Source are highlighted in green . The best and second-best adaptation approaches for each model scale are indicated in **bold** and underlined, respectively.

| | Approach | MT | | | | | SUM | | | | | MRC | | | | | MMLU | | | | |
|---|---|---|---|---|---|---|---|---|---|---|---|---|---|---|---|---|---|---|---|---|---|
| | | ne | ky | am | ha | ig | ne | ky | am | ha | ig | ne | ky | am | ha | ig | ne | ky | am | ha | ig |
| | Source | 27.0 | 21.1 | 5.1 | 24.4 | 23.0 | 22.4 | 22.9 | 8.6 | 23.7 | 23.3 | .382 | .379 | .276 | .332 | .301 | .301 | .301 | .276 | .321 | .323 |
| 7B | FFT | 32.5 | **33.8** | **12.1** | 38.6 | 36.7 | 22.1 | 23.7 | 9.3 | 32.2 | 26.4 | .360 | .441 | .309 | .460 | .396 | .293 | .312 | .288 | **.372** | .360 |
| | AdaLoRA | 28.1 | 22.3 | 4.0 | 22.9 | 22.3 | 21.7 | 23.1 | 6.5 | 31.6 | **26.6** | .351 | .343 | .276 | .328 | .291 | .309 | .311 | .272 | .278 | .324 |
| | HFT | 32.7 | 32.4 | 9.6 | 37.5 | 36.9 | 22.4 | 23.8 | 8.6 | 32.1 | 26.3 | .368 | .411 | .282 | .438 | .388 | .293 | .314 | .287 | .346 | **.373** |
| | GMT | 32.3 | 33.5 | 11.6 | 39.0 | 38.3 | 22.3 | 23.8 | 9.9 | 32.4 | 26.2 | .346 | .419 | .312 | .451 | .398 | .279 | .308 | **.296** | .353 | .361 |
| | SSU-Rand | 33.2 | 32.6 | 9.5 | 38.4 | 37.3 | 22.4 | 23.8 | 8.8 | 32.2 | 26.4 | .388 | .428 | .299 | .457 | .401 | .305 | .311 | .288 | .362 | .355 |
| | SSU-Mag | 33.1 | 32.2 | 9.7 | 37.1 | 36.6 | 22.2 | 23.7 | 9.2 | 32.3 | 26.2 | .372 | .418 | .297 | .451 | .379 | .303 | .307 | .291 | .346 | .348 |
| | SSU-Wanda | 34.0 | 32.2 | 9.0 | 42.6 | 37.1 | 22.4 | 24.2 | 8.9 | 32.2 | 26.3 | .401 | .458 | .316 | .439 | .401 | .313 | .329 | .296 | .355 | .371 |
| | Source | 32.4 | 22.5 | 6.0 | 25.3 | 25.7 | 22.9 | 23.2 | 10.0 | 25.3 | 22.4 | .501 | .393 | .318 | .348 | .310 | .345 | .322 | .293 | .333 | .351 |
| 13B | FFT | 37.5 | **36.9** | **16.5** | 40.2 | 37.1 | 21.8 | 23.7 | 10.6 | 32.7 | 25.4 | .500 | **.564** | **.381** | **.579** | .438 | .342 | .335 | .315 | **.417** | **.397** |
| | AdaLoRA | 33.7 | 24.0 | 5.7 | 26.3 | 25.4 | 22.2 | 22.9 | 9.4 | 31.6 | 25.4 | .448 | .391 | .293 | .371 | .322 | .340 | .307 | .277 | .324 | .307 |
| | HFT | 37.6 | 36.3 | 14.4 | 41.6 | 38.4 | 21.9 | 23.4 | 10.4 | 32.4 | 26.1 | .498 | .538 | .376 | .538 | .429 | .348 | .356 | .312 | .384 | .375 |
| | GMT | 37.3 | 36.6 | 16.5 | 40.2 | 36.8 | 22.0 | 23.4 | 9.8 | 32.7 | 26.0 | .501 | .559 | .355 | .530 | .420 | .348 | .356 | .318 | .404 | .338 |
| | SSU-Rand | 37.5 | 36.1 | 14.5 | 41.8 | 37.9 | 22.3 | 23.4 | 10.4 | 32.9 | 26.1 | .492 | .556 | .364 | .540 | .440 | .352 | .361 | .313 | .383 | .369 |
| | SSU-Mag | 37.2 | 36.1 | 14.5 | 39.7 | 36.5 | 22.0 | 23.0 | 9.7 | 32.1 | 26.0 | .474 | .533 | .361 | .546 | .419 | .345 | .357 | .311 | .394 | .342 |
| | SSU-Wanda | 37.9 | 35.7 | 13.7 | 44.0 | 39.1 | 22.8 | 23.8 | 11.0 | 32.3 | 25.9 | .520 | .549 | .377 | .542 | .441 | .354 | .355 | .302 | .390 | .395 |

Table 14: Performance of additional baselines: LoTA and S2FT with SSU-Wanda. We use Igbo as the target language. **Bold** and underlined denote best and second-best adaptation approaches with relative changes in subscripts. ★ indicates that the approach is a default baseline used in §6.

| Approach | Chat and Instruction-following | | | | Safety | Source language | | | | Target language (Igbo) | | | |
|---|---|---|---|---|---|---|---|---|---|---|---|---|---|
| | IFEval | AE2 | MTB | GSM8K | T3 | MT | SUM | MRC | MMLU | MT | SUM | MRC | MMLU |
| Source | $.675_{+0.0}$ | $32.6_{+0.0}$ | $3.98_{+0.0}$ | $.796_{+0.0}$ | $.851_{+0.0}$ | $28.5_{+0.0}$ | $22.8_{+0.0}$ | $.880_{+0.0}$ | $.618_{+0.0}$ | $23.0_{+0.0}$ | $23.3_{+0.0}$ | $.301_{+0.0}$ | $.323_{+0.0}$ |
| SSU-Wanda | $.670_{-0.7}$ | $25.0_{-23.2}$ | $3.92_{-1.5}$ | $\mathbf{.756}_{-5.0}$ | $\mathbf{.851}_{-0.0}$ | $46.3_{+62.7}$ | $\mathbf{23.3}_{+2.3}$ | $\mathbf{.870}_{-1.1}$ | $.603_{-2.4}$ | $37.1_{+61.7}$ | $26.3_{+12.9}$ | $\underline{.401}_{+33.2}$ | $.371_{+14.9}$ |
| LoTA (12.5%) | $.367_{-45.6}$ | $5.4_{-83.4}$ | $3.10_{-22.1}$ | $.590_{-25.9}$ | $.811_{-4.7}$ | $42.1_{+47.9}$ | $20.4_{-10.4}$ | $.857_{-2.6}$ | $.587_{-5.0}$ | $37.1_{+61.7}$ | $26.3_{+12.9}$ | $\mathbf{.402}_{+33.5}$ | $\mathbf{.374}_{+15.8}$ |
| LoTA (25.0%) | $.366_{-45.8}$ | $5.0_{-84.6}$ | $3.09_{-22.3}$ | $.590_{-25.9}$ | $.812_{-4.6}$ | $42.2_{+48.3}$ | $20.4_{-10.4}$ | $.857_{-2.6}$ | $.587_{-5.0}$ | $37.1_{+61.7}$ | $\underline{26.4}_{+13.4}$ | $\mathbf{.402}_{+33.5}$ | $\mathbf{.374}_{+15.8}$ |
| LoTA (37.5%) | $.367_{-45.6}$ | $4.9_{-85.0}$ | $3.02_{-24.1}$ | $.590_{-25.9}$ | $.811_{-4.7}$ | $42.5_{+49.3}$ | $20.4_{-10.4}$ | $.857_{-2.6}$ | $.587_{-5.0}$ | $37.2_{+62.1}$ | $\mathbf{26.5}_{+13.8}$ | $\mathbf{.402}_{+33.5}$ | $\mathbf{.374}_{+15.8}$ |
| LoTA (50.0%) | $.449_{-33.4}$ | $8.3_{-74.5}$ | $3.45_{-13.3}$ | $.636_{-20.1}$ | $.824_{-3.2}$ | $45.8_{+60.9}$ | $21.5_{-5.6}$ | $.844_{-4.1}$ | $.590_{-4.6}$ | $\underline{37.8}_{+64.7}$ | $\underline{26.4}_{+13.4}$ | $\mathbf{.402}_{+33.5}$ | $\underline{.372}_{+15.2}$ |
| LoTA (62.5%) | $.508_{-24.7}$ | $8.8_{-73.0}$ | $3.49_{-12.3}$ | $.660_{-17.1}$ | $.832_{-2.3}$ | $\mathbf{46.7}_{+64.1}$ | $21.6_{-5.1}$ | $.853_{-3.1}$ | $.596_{-3.6}$ | $\mathbf{37.9}_{+65.1}$ | $\underline{26.4}_{+13.4}$ | $\mathbf{.402}_{+33.5}$ | $.370_{+14.6}$ |
| LoTA (75.0%) | $.573_{-15.1}$ | $10.2_{-68.7}$ | $3.76_{-5.5}$ | $.672_{-15.6}$ | $.838_{-1.6}$ | $46.3_{+62.7}$ | $22.2_{-2.5}$ | $.853_{-3.1}$ | $.593_{-4.1}$ | $37.6_{+63.8}$ | $26.3_{+12.9}$ | $.389_{+29.2}$ | $.369_{+14.3}$ |
| LoTA (87.5%) | $.648_{-4.0}$ | $18.0_{-44.7}$ | $3.84_{-3.5}$ | $.681_{-14.5}$ | $.844_{-0.8}$ | $45.8_{+60.9}$ | $\underline{22.9}_{+0.6}$ | $.863_{-1.9}$ | $\underline{.603}_{-2.4}$ | $35.1_{+52.9}$ | $26.2_{+12.5}$ | $.376_{+24.9}$ | $.348_{+7.8}$ |
| ★ LoTA (90%) | $.638_{-5.4}$ | $20.4_{-37.4}$ | $\underline{3.98}_{+0.0}$ | $.706_{-11.3}$ | $.827_{-2.8}$ | $45.2_{+58.8}$ | $22.7_{-0.3}$ | $\underline{.864}_{-1.8}$ | $\mathbf{.606}_{-2.0}$ | $34.4_{+49.9}$ | $26.2_{+12.5}$ | $.366_{+21.5}$ | $.360_{+11.5}$ |
| ★ S2FT (Down) | $\mathbf{.695}_{+3.0}$ | $\mathbf{27.9}_{-14.3}$ | $\mathbf{3.99}_{+0.3}$ | $.732_{-8.0}$ | $.834_{-2.0}$ | $36.7_{+29.0}$ | $22.6_{-0.7}$ | $.857_{-2.6}$ | $\underline{.603}_{-2.4}$ | $21.7_{-5.4}$ | $26.0_{+11.6}$ | $.303_{+0.6}$ | $.331_{+2.5}$ |
| S2FT (Down + Output) | $.635_{-5.9}$ | $19.5_{-40.1}$ | $3.75_{-5.7}$ | $.306_{-61.6}$ | $.822_{-3.4}$ | $30.0_{+5.4}$ | $21.9_{-3.8}$ | $.632_{-28.2}$ | $.393_{-36.4}$ | $19.7_{-14.2}$ | $25.3_{+8.6}$ | $.279_{-7.3}$ | $.245_{-24.1}$ |
| S2FT (Down; $r = 16$) | $\underline{.678}_{+0.5}$ | $\underline{25.7}_{-21.1}$ | $3.96_{-0.5}$ | $\underline{.735}_{-7.7}$ | $.841_{-1.2}$ | $38.7_{+36.0}$ | $22.8_{+0.1}$ | $.852_{-3.2}$ | $\mathbf{.606}_{-2.0}$ | $24.7_{+7.6}$ | $25.9_{+11.2}$ | $.314_{+4.3}$ | $.328_{+1.6}$ |
| S2FT (Down; $r = 32$) | $.661_{-2.0}$ | $21.6_{-33.7}$ | $3.92_{-1.5}$ | $.706_{-11.3}$ | $.837_{-1.7}$ | $41.7_{+46.5}$ | $22.7_{-0.3}$ | $.860_{-2.3}$ | $\underline{.603}_{-2.4}$ | $27.4_{+19.4}$ | $26.1_{+12.1}$ | $.316_{+4.9}$ | $.333_{+3.1}$ |
| S2FT (Down; $r = 64$) | $.652_{-3.4}$ | $19.7_{-39.5}$ | $3.82_{-4.0}$ | $.683_{-14.2}$ | $\underline{.846}_{-0.6}$ | $43.2_{+51.8}$ | $\underline{22.9}_{+0.6}$ | $.859_{-2.4}$ | $\underline{.603}_{-2.4}$ | $31.0_{+35.1}$ | $26.3_{+12.9}$ | $.317_{+5.3}$ | $.344_{+6.5}$ |

# D  SUPPLEMENTARY ANALYSIS

In §6, we use the default configurations for additional baselines: LoTA and S2FT. To ensure a comprehensive evaluation, we extend this with a fine-grained hyperparameter ablation study (Table 14).

**LoTA.**  We examine LoTA across sparsity ratios in 12.5% increments, consistent with our analysis of SSU, HFT, and GMT. High sparsity ratios (e.g., 90% and 87.5%) preserve source performance reasonably well while improving target performance. Despite these gains, these configurations consistently underperform SSU-Wanda. At 90% sparsity, LoTA shows lower target gains (e.g., 23.9% relative average gain vs. 30.7% for SSU-Wanda) and weaker source preservation (e.g., 7.6% average drop in monolingual source tasks vs. 4.0%). Conversely, lower sparsity allows for more adaptation and leads to better target performance. For instance, LoTA at 50% achieves a 31.7% average target gain, surpassing the 30.7% gain of SSU-Wanda. However, this improvement triggers substantial catastrophic forgetting: the average drop in monolingual source tasks reaches 19.9%, substantially worse than the 7.6% drop at 90% sparsity. This degradation intensifies at 37.5% sparsity, reaching a 25.4% drop. These results indicate that while the default high-sparsity setting mitigates catastrophic forgetting in LoTA, the approach fails to match the balance of source preservation and target language acquisition achieved by SSU-Wanda.

**S2FT.**  Following the original paper (Yang et al., 2024), we sparsely tune the down projection layers using a parameter count equivalent to LoRA with a rank of 8 (Table 8). We additionally evaluate larger parameter budgets equivalent to ranks of 16, 32, and 64. We also test the combination of "Down and Output" projection tuning to determine if the poor performance reported for Mistral and Llama3 (attributed to inflexible selection in multi-query attention) applies to OLMo 2.

First, as noted in §6, the default setting preserves source capabilities effectively (3.3% average drop vs. 4.0% for SSU-Wanda) but yields minimal target gains (2.3% vs. 30.7%). Increasing the trainable parameter budget (i.e., reducing sparsity) improves target performance but erodes source capabilities. At the equivalent of rank 64, S2FT exhibits a larger source drop (8.2%) than SSU-Wanda (4.0%) while still achieving lower target gains (15.0% vs. 30.7%). As larger capacities progressively degrade source performance without matching the target gains of SSU-Wanda, we conclude that no optimal S2FT configuration exists to surpass SSU in our problem setup. Finally, we confirm that tuning "Down and Output" projections yields suboptimal results, causing severe relative drops of up to 23.1% in monolingual source tasks and 9.25% in target tasks. In summary, regardless of hyperparameter adjustments, only SSU provides robust source preservation while elevating target language abilities to levels comparable to FFT.

# E    EXTENDED RELATED WORK

SSU addresses the core challenge of continual learning (CL): adapting a model to new tasks while mitigating catastrophic forgetting (Goodfellow et al., 2015; Kirkpatrick et al., 2017). This section situates SSU within the parameter-centric family of CL solutions. These methods protect knowledge at the parameter level, typically without accessing data from the old task for replay. They generally address two fundamental questions: (1) the **Identification Problem**, defining which parameters are critical to a previous task; and (2) the **Protection Problem**, determining the mechanism to enforce protection on those parameters. Parameter-centric approaches largely fall into three categories: **soft, regularization-based** protection; **hard, architectural-based** protection; and **adaptive, hybrid** methods.

**Soft Parameter Protection (Regularization-Based).**    These methods discourage changes to critical parameters by adding a penalty term to the loss function of the new task. Approaches differ primarily in solving the "Identification Problem." Elastic Weight Consolidation (EWC) identifies critical parameters via the Fisher Information Matrix diagonal (Kirkpatrick et al., 2017), while Synaptic Intelligence (SI) computes importance online by tracking the cumulative contribution of each parameter to loss reduction (Zenke et al., 2017). Similarly, Memory Aware Synapses (MAS) estimates importance weights based on the sensitivity of the learned function (output function) to parameter changes, eliminating the need for original labeled data (Aljundi et al., 2018). Soft-Masking of Parameter-Level Gradient Flow (SPG) protects knowledge by directly modulating gradient flow with soft masks rather than modifying the loss objective (Konishi et al., 2023). However, such "soft" constraints often fail under severe distributional shifts (Wang et al., 2023). This limitation becomes particularly acute in our problem setup (i.e., adapting instruct models using unlabeled target language data), where optimization pressure from unlabeled target corpora can overpower regularization penalties.

**Hard Parameter Protection (Isolation & Architectural).**    These methods enforce stability via structural constraints, such as freezing or allocating parameters, to ensure near-zero forgetting. Hard Attention to the Task (HAT) learns a binary mask, forcing gradients to zero for parameters allocated by the mask from any previous task (Serra et al., 2018). PackNet employs an "iterative prune, fix, and retrain" cycle, freezing the surviving "packed" weights and forcing new tasks to utilize only "free" parameters (Mallya & Lazebnik, 2018). Piggyback represents an extreme form, freezing an entire pre-trained backbone and learning new tasks solely by training new binary masks (Mallya et al., 2018).

**Adaptive & Hybrid Protection.**    This emerging class assesses the properties of an incoming task to select a protection strategy dynamically. Context-aware Task-driven (CAT) automatically detects whether a new task resembles previous ones (Ke et al., 2020), applying Hard Protection (binary mask) for dissimilar tasks and Soft Protection (attention) for similar tasks. Parameter Allocation & Regularization (PAR) identifies task relatedness and applies dynamic protection: "easy" tasks are consolidated via soft regularization, while "difficult" tasks trigger the hard allocation of a new, isolated expert model (Wang et al., 2023). While promising, the application of such dynamic allocation strategies to the specific constraints of LLM language adaptation remains an interesting avenue for future research.

**Situating SSU within Continual Learning.**    SSU adapts these CL principles for the linguistic adaptation of instruct LLMs. We characterize SSU as a source-focused method utilizing static hard parameter protection. It resolves the "Identification Problem" via source-data-driven importance scores (e.g., Wanda) and the "Protection Problem" via column-wise structural freezing. While conceptually aligned with hard protection, SSU overcomes specific limitations regarding **problem setting** and **scale**. Foundational CL methods largely focus on task-incremental learning, where the model learns a sequence of discrete, labeled tasks (e.g., Task 1: MNIST, Task 2: CIFAR). Consequently, methods like HAT rely on task identifiers (Task IDs) at inference time to select the correct mask. This requirement is incompatible with general-purpose instruct LLMs, where the input language (or task) is unknown and the model must operate as a unified entity without external task signals. Regarding scale, foundational methods typically target architectures with fewer than 1B parameters (e.g., PackNet uses VGG-16 ($\sim$138M) (Simonyan & Zisserman, 2015)). Methods like the iterative pruning and retraining cycles of PackNet often become computationally prohibitive when applied to billion-parameter LLMs. In contrast, SSU utilizes a one-shot, static calculation of importance before training, making it computationally viable for modern transformer-based architectures.

