# OpenReview forum: "Mitigating Catastrophic Forgetting in Target Language Adaptation of LLMs via Source-Shielded Updates"
_ICLR.cc/2026/Conference — ICLR 2026 Conference Withdrawn Submission_

### Official Review · Reviewer_A7vK · 2025-10-29

**Soundness:** 3
**Presentation:** 4
**Contribution:** 3
**Rating:** 8
**Confidence:** 4

**Summary:**

This paper focuses on adapting instruct-tuned LLMs to target languages using only unlabled target language text. Standard continual pretraining on target language data often results in catastophic forgetting where the new training data erases source knowledge and significnatly affects the core chat and instruction-following capabilities. To address this issue, this paper proposes Source-Shielded Updates, a simple and effective source-focused appraoch that shields source knowledge by freezing specific columns of the weight matrices while training on target language data. The columns to be frozen are selected by using a small set of source data and a parameter importance scoring method.

The proposed approach is verified by adapting 7B and 13B OLMo2 instruct models (trained on English-dominated Common Crawl corpus) to five different target languages. Experimental results demonstrate that SSU consistently outperforms relevant baselines in terms of target-language proficiency while preserving general source-language performance.

**Strengths:**

The proposed approach is simple and easy to use, which I consider as strengths given its effectiveness.

Different from existing approaches that use target data-driven signals to identify which parameters are to be trained, the proposed approach is source-focused and uses source data to select parameters that are kept frozen during training. This makes sense and also works better compared to target data-driven GMT method.

Several downstream evaluation datasets have been used to demonstrate source knowledge preservation.

Ablation studies were conducted demonstrating the effect of freezing ratio on the trade-off between source knowledge retention and target knowledge acquisition.

Effectiveness of proposed SSU strategy is demonstrated using multiple importance scoring methods.

**Weaknesses:**

The proposed approach is compared with alternative selection straggles at 50% freezing ratio. While 50% provides a good operating point for the proposed approach (in terms of trade-off between source and target metrics), it may not be the best operating point for the alternative approaches. So, the current results do not provide a full comparison between different selective update strategies.

**Questions:**

Getting plots like Fig.2 for the other selective update strategies will make the comparison between different approaches more thorough.

---

> ### Author Response · Authors · 2025-11-12
>
> Many thanks for your constructive and positive feedback. We are thrilled that you found the paper clear and the approach effective, and we appreciate you highlighting its strengths.
>
> Regarding the identified weakness, we acknowledge that the freezing ratio is a key hyperparameter, and that different methods might have different optimal ratios. To address this, we are currently conducting the exact analysis you suggested. We are generating plots like Figure 2 for our key baselines (HFT and GMT) on the OLMo-2 7B / Igbo setting. We will update the paper with the corresponding results as soon as they are available, well within the response period.
>
> We would also like to kindly note that our original setup (L227-230) intentionally followed the precedent set by Hui et al. (ACL 2025), using their default 50% freezing ratio for HFT to ensure a fair comparison with the original work.
>
> We will post a summary of our findings here in the forum to help with easier understanding once the experiments are complete.
>
> Thank you again for your valuable suggestions for strengthening the paper.

---

> > ### Author Response · Authors · 2025-11-20
> >
> > Dear Reviewer A7vK,
> >
> > Thanks again for the constructive and positive feedback, and for recognizing the simplicity, effectiveness, and strong empirical results of the SSU framework.
> >
> > We have completed the requested analysis (i.e., plotting the performance of our key baselines across different freezing ratios):
> >
> > * **Expanded Hyperparameter Analysis:** A fine-grained ablation study of the freezing ratio for HFT and GMT was conducted, and the results were integrated into the existing Figure 2.
> > * **Finding:** We find that SSU consistently outperforms HFT across all ratios, and offers superior source retention than GMT regardless of the ratio. This confirms SSU as the optimal method for simultaneously achieving strong source preservation and high target language gains.
> >
> > This addition provides a comprehensive comparison between the selective update strategies, addressing the concern. We welcome any further questions.
> >
> > Sincerely,
> >
> > The Authors of Submission 9371

---

### Official Review · Reviewer_7cj2 · 2025-10-31

**Soundness:** 3
**Presentation:** 3
**Contribution:** 2
**Rating:** 4
**Confidence:** 3

**Summary:**

The paper introduces Source-Shielded Updates (SSU), a parameter update strategy to mitigate catastrophic forgetting during target language adaptation of instruction-tuned LLMs. SSU first identify critical parameters with Wanda technique, then freeze the corresponding parameters to prevent gradient update. Experiments conducted on five low-resource languages and two model scales (7B, 13B) demonstrate that SSU reduces catastrophic forgetting by a substantial margin compared to full fine-tuning and several strong baselines.

**Strengths:**

The question of adapting an instruction tuned LLM to support new languages without forgetting is critical.
The proposed method is well motivated and mathematically sound.
The empirical results are strong. The proposed method successfully reduced forgetting.

**Weaknesses:**

1.	The novelty is largely overclaimed and many related works and baselines are not included as they should be. First, the idea of freezing critically parameters for learned knowledge has been widely adopted. A classical CL approaches, HAT, shares this idea. More recent works such as CAT and SPG. Therefore, the claim that `However, existing paradigms are ill-suited for the specific challenge of adapting instruct models with unlabeled target language text. They either rely on random selection, offering no principled way to preserve knowledge, or on signals from the new data to guide updates (target-focused).` is not true. Additionally, many CL baselines are missing.

2.	The evaluation mostly considers ICL ability of LLMs. It would be better to include more generation/language modeling tasks to eval the new language ability.

3.	New language typically faces tokenizer issue. For example, new language may be OOV or be over tokenized. This requires updating the dictionary. The current approach do not support this.

**Questions:**

See above.

---

> ### Author Response · Authors · 2025-11-13
>
> Thank you for your constructive review and for acknowledging the paper's clarity and soundness. Below, we provide a point-to-point response to each of the weaknesses raised.
>
> ---
> **Point 1A: Novelty is overclaimed; related work (HAT [1], CAT [2], SPG [3]) is missing.**  Thank you for highlighting this continual learning (CL) literature. We agree that methods like [1]-[3] share the core principle of protecting critical parameters. We will revise Related Work to include and discuss these methods. Our revised discussion will also clarify the key distinction motivating our work: these foundational methods were primarily designed for task-incremental learning (e.g., sequential classification on smaller models with up to \~50M parameters). In contrast, SSU addresses the linguistic adaptation of billion-parameter LLMs, a distinct challenge in terms of objective, scale, and computational complexity. We will situate SSU within this broader context to clarify its novelty for this specific problem.
>
> [1] Overcoming catastrophic forgetting with hard attention to the task (Serrà et al., ICML 2018)
> [2] Continual Learning of a Mixed Sequence of Similar and Dissimilar Tasks (Ke et al., NeurIPS 2020)
> [3] Parameter-Level Soft-Masking for Continual Learning (Konishi et al., ICML 2023)
>
> ---
> **Point 1B: A specific claim ("existing paradigms are ill-suited...") is "not true."** This is an excellent point. We apologize for the imprecise language. Our claim was intended to *differentiate SSU from other selective parameter-tuning strategies for LLM adaptation*, not to dismiss the entire field of CL. We will revise this sentence (L56-59) to be more precise: "*However, existing strategies **within selective parameter tuning for adapting LLMs** are ill-suited...*"
>
> ---
> **Point 1C: Many CL baselines are missing.** Our rationale for the original baseline selection was to compare SSU against the most direct state-of-the-art methods for our specific problem: linguistic adaptation of large-scale LLMs. This is why we selected strong, relevant baselines such as HFT and GMT, both proposed for LLM adaptation.
>
> As discussed in Point 1A, applying classical CL methods like HAT (which experimented on models up to 7.1M parameters) to billion-parameter models is a non-trivial research and engineering challenge. Their core mechanics (e.g., calculating and storing per-parameter masks) can be computationally infeasible at this scale.
>
> However, to strengthen our empirical validation, we agree that including more methods that mitigate forgetting is essential. Therefore, we will broaden our comparison by adding experiments for two additional, highly relevant LLM-specific baselines (i, ii), as suggested by Reviewer gbcc. These methods provide a stronger, more direct comparison. We will also note that investigating conventional CL methods for LLM adaptation is an important future work.
>
> (i) Lottery Ticket Adaptation: Mitigating Destructive Interference in LLMs (Panda et al., 2024)
> (ii) S2FT: Efficient, Scalable and Generalizable LLM Fine-tuning by Structured Sparsity (Yang et al., NeurIPS 2024)
>
> ---
> **Point 2: On the width of the evaluation.** We appreciate the suggestion. We would like to kindly clarify that our evaluation already includes both generation and classification tasks to evaluate target language ability. Specifically:
> * Generation: We use summarization and machine translation.
> * Classification: We use Belebele and Global MMLU.
>
> This evaluation protocol follows that of Yamaguchi et al. (TMLR 2025) (1). Regarding language modeling, we respectfully note that because our base models are instruction-tuned, standard perplexity or LM tasks are not the primary focus. Our goal is to preserve source abilities while adapting to a target language, which we believe our current task suite captures well.
>
> (1) Adapting Chat Language Models Using Only Target Unlabeled Language Data.
>
> ---
> **Point 3: On tokenizer adaptation.** We clarify that the models used in our experiments, like modern LLMs, use a byte-level BPE tokenizer. This design choice fundamentally prevents any out-of-vocabulary issues, which is the most critical part of the challenge you have raised.
>
> The remaining issue is tokenization efficiency (i.e., over-fragmentation). While we agree this is important for optimal performance, we view tokenizer adaptation as a separate, pre-processing step that is orthogonal to our paper's contribution.
>
> The scope of this work is to investigate catastrophic forgetting during parameter updates, assuming a fixed model architecture. To make this scope explicit, we will add a footnote clarifying that while tokenizer adaptation is out of scope for this study, we view it as a complementary and important research direction. We will also note that combining parameter update strategies (like SSU) with tokenizer adaptation is a promising avenue for future work.
>
> Thank you again for your suggestions. We will incorporate these clarifications into the final paper.

---

> ### Author Response · Authors · 2025-11-20
>
> Dear Reviewer 7cj2,
>
> Thank you for your constructive review, for acknowledging the critical nature of the problem, and for recognizing the method's motivation and empirical strength.
>
> We have now incorporated revisions to address the concerns regarding the novelty claim, baselines, and textual clarity. The changes are detailed in the manuscript (highlighted in blue), and a summary is provided below.
>
> | Concern in the Review | Action Taken & Location in Paper | Key Finding / Note |
> | :--- | :--- | :--- |
> | **Novelty is overclaimed; key CL works (HAT, CAT, SPG) are missing.** | Extended the Related Work discussion (Footnote 3 & Appendix E) to situate SSU within the parameter-centric family of CL solutions. | We clarify that SSU is distinct due to its focus on large-scale LLM linguistic adaptation using a proactive, column-wise structural mask, a different objective and scale than foundational CL methods. |
> | **Claim that "existing paradigms are ill-suited..." is too broad/untrue.** | The claim in L56-59 was refined. | The revised text now accurately focuses on differentiating SSU from other selective parameter-tuning strategies for adapting LLMs (e.g., random or target-focused methods). |
> | **Missing CL baselines / need for more baselines.** | We added empirical comparisons against two highly relevant LLM-specific baselines: LoTA and S2FT. (Table 6) | We confirm that SSU offers a superior Pareto frontier compared to both LoTA and S2FT. |
> | **Tokenizer issues are unaddressed.** | A clarification was added (Footnote 2). | We clarified that tokenization efficiency is an important, orthogonal direction for future work, separate from our paper's scope of selective parameter update strategy. |
> | **Insufficient generation/language modeling tasks.** | (No change made to the evaluation suite.) | We reiterate that our evaluation already includes both generation (summarization, machine translation) and classification tasks, providing a comprehensive assessment of the target language adaptation ability, consistent with related literature. |
>
> We believe these comprehensive revisions, including expanded empirical comparisons, additional discussion of the CL literature, and the refinement of our claim, fully address your concerns and strengthen the contributions of the paper.
>
> We appreciate your guidance and welcome any further questions or feedback.
>
> Sincerely,
>
> The Authors of Submission 9371

---

### Official Review · Reviewer_t7Au · 2025-11-01

**Soundness:** 2
**Presentation:** 3
**Contribution:** 2
**Rating:** 2
**Confidence:** 4

**Summary:**

The paper introduces Source-Shielded Updates (SSU), a source-driven selective-parameter update framework for adapting instruct-tuned LLMs to underrepresented languages using only unlabeled target-language text. SSU aims to mitigate catastrophic forgetting while retaining source-language instruction-following abilities.

**Strengths:**

1. The writing is clear, well-structured, and easy to follow, making the paper accessible and logically organized.


2. The column-wise masking design is an elegant and technically sound insight, offering a simple yet effective structural approach to preserve model representations.


3. The source-driven importance scoring provides a principled and data-grounded alternative to random or target-data-based freezing strategies.

**Weaknesses:**

1. The paper lacks a comprehensive hyperparameter search for baseline methods, which may make the reported comparisons less fair or less reproducible.


2. The proposed method benefits from additional source-language data for parameter importance estimation, while baseline methods do not, introducing a potential source of unfair advantage.


3. The novelty is limited, as similar importance-based freezing or selective update methods have been explored in prior works [1,2].


4. The paper does not include baselines that also leverage source data, which would provide a more balanced evaluation of the benefits of source-informed adaptation.


5. The comparison omits recent state-of-the-art methods addressing catastrophic forgetting in multilingual CPT, making it difficult to precisely quantify how much SSU advances the field.

[1] Jung, S., Ahn, H., Cha, S., & Moon, T. (2020). Continual Learning with Node-Importance based Adaptive Group Sparse Regularization.

[2] Yao, K., Gao, P., Li, L., Zhao, Y., Wang, X., Wang, W., & Zhu, J. (2024). Layer-wise Importance Matters: Less Memory for Better Performance in Parameter-efficient Fine-tuning of Large Language Models.

**Questions:**

Would it be possible to modify existing baselines or introduce new ones that also incorporate source data to ensure a more fair comparison?

---

> ### Author Response · Authors · 2025-11-13
>
> Thank you for your detailed and constructive feedback. We appreciate your positive comments on the paper's clarity and the design of SSU. We offer the following point-by-point responses to the weaknesses you identified:
>
> ---
> **On the hyperparameters for baselines**: We agree that the freezing ratio is a key hyperparameter. To address this, we are conducting an analysis for our key baselines (HFT and GMT) on the OLMo-2 7B / Igbo setting, similar to Figure 2 in the paper. We will add this to the paper to ensure a fair and comprehensive comparison.
>
> We would also like to kindly note that our original setup (L227-230) intentionally followed the precedent set by Hui et al. (ACL 2025), using their default 50% freezing ratio for HFT to ensure a fair comparison with the original work. Moreover, our code is publicly available on the anonymous GitHub repository. Therefore, we do not think this relates to reproducibility issues.
>
> ---
> **On the fairness of using calibration data**: We acknowledge that SSU uses a small set of source samples (500 examples) for calibration. To quantify any potential advantage, we ran a new analysis with an even smaller set (128 examples, \~0.26M tokens), and found a maximum performance difference of only 1.2 points (please see the table below).
>
> Table: Performance of SSU-Wanda with different number of calibration samples.
> We use Igbo as the target language and tulu-3-sft-olmo-2-mixture as the calibration dataset.
> **Bold** indicates best adaptation approach.
> |Approach|IFEval|AE2|MTB|GSM8K|T3|en MT|en SUM|en MRC|en MMLU|ig MT|ig SUM|ig MRC|ig MMLU|
> |:-|:-:|:-:|:-:|:-:|:-:|:-:|:-:|:-:|:-:|:-:|:-:|:-:|:-:|
> |Source|.675|32.6|3.98|.796|.851|28.5|22.8|.880|.618|23.0|23.3|.301|.323|
> |500 examples|.670|**25.0**|**3.92**|**.756**|.851|46.3|**23.3**|.870|**.603**|37.1|**26.3**|.401|**.371**|
> |128 examples|**.682**|24.3|3.89|.754|**.852**|**46.4**|23.2|**.873**|.600|**37.2**|**26.3**|**.410**|**.371**|
>
> We believe this shows the setup is not unfair, for two reasons:
> * The performance is not sensitive to this small amount of data.
> * The 500 examples (\~1M tokens) or 128 examples (\~0.26M) are negligible compared to the 200M-token adaptation corpus.
>
> We will add this new analysis to the paper to demonstrate this.
>
> ---
> **On the novelty of the proposed method**: Thank you for pointing out [1] and [2]; we will additionally cite and discuss them in our revision. However, we would like to highlight key distinctions that establish SSU's novelty. For instance:
>
> [1] Jung et al. (2020) propose an online regularization method. It is a reactive approach that penalizes changes to important weights during a continual learning process. In contrast, SSU is a proactive, selective-update framework. It identifies and freezes structurally important parameters (columns) before adaptation begins, specifically for the "one-shot" domain adaptation to a new language with unlabeled text, which is a different setting from [1].
>
> [2] Yao et al. (2024) focus on Parameter-Efficient Fine-Tuning (PEFT), specifically determining which layers are best for inserting adapters (e.g., LoRA). Our work: SSU is not about PEFT. It operates on the full model parameters and introduces a column-wise structural mask to preserve capabilities. This "column-wise" structural insight, driven by source-language importance, is fundamentally different from the "layer-wise" PEFT-placement problem addressed by [2].
>
> To this end, we believe SSU's contribution, a proactive, source-driven, structural (column-wise) mask for full-parameter selective updates, is distinct and novel within this line of work. We will clarify this in our related work section.
>
> ---
> **On adding source-data-focused baselines**: This is a great point. Indeed, we have already provided two ablation baselines (see Table 3: row-wise and element-wise freezing results). We found that naively using source-data information, such as element-wise freezing, along with row-wise freezing, does not effectively retain the model's source capabilities.

---

> > ### Author Response · Authors · 2025-11-13
> >
> > ---
> > **On SOTA catastrophic forgetting methods in multilingual CPT**: We acknowledge there are several methods proposed for mitigating catastrophic forgetting in the context of multilingual CPT for LLMs (e.g., (i), (ii), (iii), (iv)). We believe most of these methods are orthogonal to our work. As we state in L125-126, SSU could likely be combined with these other methods if necessary. Our paper's contribution is within the selective-update space, and HFT/GMT are the most direct competitors in this specific area. We will emphasize this distinction more clearly in the revision, while citing relevant multilingual CPT papers in the related work section.
> >
> > (i) https://aclanthology.org/2023.findings-acl.48/ (Controlling learning rates, one of the pioneering studies in catastrophic forgetting mitigation in multilingual CPT for LLMs)
> > (ii) https://aclanthology.org/2024.findings-emnlp.1000/ (Uses iterative model merging)
> > (iii) https://aclanthology.org/2024.emnlp-main.604/ (Uses multiple model merging)
> > (iv) https://arxiv.org/abs/2509.11414v1 (Concurrent work. Selectively add LoRA modules into chosen transformer layers in an LM)
> >
> > ---
> > **Our Plan**: We are currently running the new experiments for the baseline hyperparameter analysis. We will post a summary of our findings here in the forum as soon as they are complete, and we will incorporate all results and clarifications into the final paper.
> >
> > Thank you again for helping us improve the paper.

---

> ### Author Response · Authors · 2025-11-20
>
> Dear Reviewer t7Au,
>
> Thank you again for your review, and for recognizing the clarity, structural design, and data-grounded nature of the SSU framework.
>
> We have completed the requested additional analyses and have integrated the results into the revised manuscript, directly addressing the key concerns regarding the fairness and comprehensive baseline analysis.
>
> Below, we summarize the key revisions integrated into the paper using the following table.
>
> | Concern in the Review | Action Taken & Location in Paper | Key Finding / Note |
> | :--- | :--- | :--- |
> | **Lack of comprehensive hyperparameter search for baselines.** | A fine-grained ablation study of the freezing ratio for key baselines, HFT and GMT, the same as SSU, was conducted. (Figure 2) | We confirm that SSU consistently outperforms HFT across all ratios, and offers superior source retention than GMT regardless of the ratio. |
> | **Unfair advantage from using source calibration data.** | SSU's performance sensitivity was analyzed by reducing the calibration set from the default 500 examples to a very small 128 examples (~0.26M tokens). (Table 5) | We confirm that a minimal sample size is sufficient. This supports the conclusion that the efficacy of SSU is not reliant on the quantity of source data. |
> | **Missing baselines that also leverage source data.** | The ineffectiveness of naive source-informed baselines (element-wise and row-wise freezing) is presented and discussed. (Table 3) | We confirm that simply leveraging source-related data (e.g., via naive element-wise freezing) does not effectively retain source capabilities, proving SSU's effectiveness stems from its structural (column-wise) protection mechanism. |
> | **Limited novelty/comparison to importance-based methods (e.g., [1, 2]).** | The Related Work section was enhanced to cite and discuss prior importance-based works (L143-148). | The discussion clarifies that SSU's proactive, source-driven, column-wise structural mask for full-parameter updates is distinct from the regularization (e.g., [1]) or Parameter-Efficient Fine-Tuning (PEFT) methods (e.g., [2]). |
> | **Missing SOTA catastrophic forgetting methods in multilingual CPT.** | The Related Work section was updated to include a clear discussion of relevant literature on mitigating catastrophic forgetting in multilingual CPT (L119-125). | The revision clarifies that SSU contributes to the parameter selective-update space, and the comparison against HFT/GMT is the most direct given the problem setup. |
>
> We believe these revisions comprehensively address the feedback, demonstrate the validity of SSU, and strengthen the scientific contribution of the paper.
>
> We appreciate the valuable guidance and welcome any further questions or feedback.
>
> Sincerely,
>
> The Authors of Submission 9371

---

### Official Review · Reviewer_gbcc · 2025-11-03

**Soundness:** 2
**Presentation:** 3
**Contribution:** 2
**Rating:** 2
**Confidence:** 4

**Summary:**

The paper aims to mitigate catastrophic forgetting in LLM adaptation by identifying and freezing a subset of the parameters during adaptation. Empirically, the proposed approach has better performance than some existing adaptation methods.

**Strengths:**

- Catastrophic forgetting is an important problem. The proposed approach is tested on models up to 13B and works well empirically. However the comparisons are limited to a small subset of relevant baselines and miss many important and relevant methods in the literature.

- The paper is well written.

**Weaknesses:**

- The baselines used in empirical comparisons are not comprehensive. The paper misses many key baselines, both in related work summary and in empirical comparisons, such as: [1, 2, 3]. It's important to see how the proposed method's performance compares with these relevant baselines.

[1] [Lottery Ticket Adaptation: Mitigating Destructive Interference in LLMs](https://arxiv.org/abs/2406.16797)

[2] [LoRI: Reducing Cross-Task Interference in Multi-Task Low-Rank Adaptation](https://arxiv.org/html/2504.07448v1)

[3] [S2FT: Efficient, Scalable and Generalizable LLM Fine-tuning by Structured Sparsity](https://arxiv.org/abs/2412.06289)

**Questions:**

Please see above.

---

> ### Author Response · Authors · 2025-11-13
>
> Thank you for your valuable feedback. We appreciate you acknowledging the paper's presentation quality.
>
> We agree that comparing SSU against [1, 3] will strengthen the paper. These baselines are interesting as they represent a different class of solution to the problem. For instance, methods like S2FT [3] focus on structural sparsity, while LoTA [1] introduces hard masks trained during the “mask calibration” phase. We believe this comparison will help readers better situate our contribution.
>
> However, we are afraid that LoRI [2] cannot be compared with other approaches, including SSU, in an apple-to-apple manner due to fundamental differences in prerequisites. LoRI is an adapter-based PEFT method designed for sequential learning and adapter merging, which assumes the ability to train a source adapter (e.g., a safety adapter) using labeled task-specific data. In contrast, the SSU framework is explicitly defined by its constraints: adaptation relies on unlabeled target language data (CPT), and it only requires access to a small set of source calibration data (e.g., 500 samples; 128 samples are even possible. Please see the response to Reviewer t7Au) solely for generating a structural freezing mask. Since SSU does not involve training or integrating a full source adapter, a fair comparison is methodologically infeasible given our problem setup.
>
> Kindly note that due to computational constraints, we are running these new experiments on the OLMo-2 7B model with Igbo as the target language, which matches the setting from our existing analysis section. We will update the paper and code repository with the full results as soon as they are available, well within the response period.
>
> Thank you again for the suggestions.

---

> > ### Author Response · Authors · 2025-11-20
> >
> > Dear Reviewer gbcc,
> >
> > Thank you once again for your constructive feedback.
> >
> > We have completed the expanded comparison against the key baselines, LoTA and S2FT, using the OLMo 2 7B / Igbo setting. The results and detailed analysis have been integrated into the revised manuscript (Table 6).
> >
> > **Key Findings from the New Comparison**
> > * The results show that the SSU method offers a superior Pareto frontier compared to both LoTA and S2FT.
> > * S2FT and LoTA exhibited a less favorable trade-off, as they either led to limited target language adaptation or resulted in greater source task forgetting.
> > * We also conducted extensive hyperparameter tuning for these new baselines to ensure a fair comparison, confirming the reported trends (details are in Appendix D).
> >
> > We believe these additions, along with the detailed discussions in the revised paper, comprehensively address your concerns and significantly strengthen the paper's contribution.
> >
> > We appreciate your valuable engagement and welcome any further questions or feedback you may have.
> >
> > Sincerely,
> >
> > The Authors of Submission 9371

---

> > > ### Comment · Reviewer_gbcc · 2025-11-25
> > >
> > > Thank you for including new comparisons. I see "using their default configurations." in the revised paper and Appendix D mentions hyperparameter tuning only for the sparsity level. Does this mean there was no other hyperparameter tuning for the baselines, e.g., learning rate, batch size?

---

> ### Author Response · Authors · 2025-11-25
>
> Thank you for the reply!
>
> For method-specific hyperparameters (e.g., sparsity), we used the configurations specified in the original LoTA and S2FT papers (as mentioned in the Appendix).
>
> For optimization hyperparameters (e.g., learning rate, batch size), we applied a unified setting across all approaches. This follows the experimental protocol established in recent literature for language adaptation [1].
>
> Exhaustive tuning for 7B/13B models is too resource-intensive to run for every baseline. It also introduces bias: if we tune one method more than another, the comparison is not fair. We stuck to a standard, proven configuration [1] to ensure an apples-to-apples comparison of the methods themselves.
>
> [1] Yamaguchi et al., "Adapting Chat Language Models Using Only Target Unlabeled Language Data", TMLR 2025.

---

### Author Response · Authors · 2025-11-14
**Summary of Revision Plan**

Dear Reviewers and Area Chair,

Thank you all for your constructive feedback. We have a clear plan to revise the paper based on your valuable comments. For your convenience, here is a summary of our revision plan:

Our main actions are:

1.  **Expanded Comparisons**: We are running new experiments to add two key baselines, LoTA and S2FT (Reviewers gbcc, 7cj2).
2.  **Hyperparameter Analysis**: We are conducting a freezing-ratio analysis for the HFT and GMT baselines, similar to Figure 2 in the paper, to ensure a fair comparison (Reviewers t7Au, A7vK).
3.  **New Analysis:** We will add an analysis showing SSU is not sensitive to the small calibration data size, addressing fairness concerns (Reviewer t7Au).
4.  **Text Revisions:** We will revise the paper to: (i) enhance the “Related Work” section to better situate SSU against continual learning and other methods, (ii) refine our claims in L56-59, and (iii) clarify that tokenizer adaptation is out of scope but an important topic for future work (Reviewers 7cj2, t7Au).

Due to computational constraints, these additional experiments are being conducted on the OLMo-2 7B / Igbo setting, consistent with our existing analysis. We are already running the new experiments and will incorporate all results, analyses, and textual revisions into the final version of the paper. We will post an update with the new results as soon as they are available within the author response period.

Thank you again for your guidance.

---

### Author Response · Authors · 2025-11-20
**Revision note**

Dear Reviewers and Area Chair,

Thank you for the continued engagement with our work. Building on our previous update, we completed the additional experiments and integrated the results into the revised manuscript. The changes are outlined below.

To facilitate the review process, all the major changes in the manuscript are in **blue**.

---
**1. On the comparison against more baselines: LoTA and S2FT** (Response to Reviewers gbcc, 7cj2)

As requested, we compare SSU against LoTA (Panda et al., 2024) and S2FT (Yang et al., 2024). The results shown in Table 6 of the revised paper highlight the superior trade-off offered by SSU. Conversely, S2FT and LoTA either suffer from limited target adaptation or greater source forgetting. The column-wise structural freezing of SSU provides a more effective Pareto frontier than the lottery-ticket or structured sparsity approaches used by these baselines.

To ensure fair comparison, we conducted extensive hyperparameter tuning for these new baselines, mirroring the analysis for SSU/HFT/GMT (Figure 2). We confirm that the trend holds across different hyperparameter settings (Please see Appendix D).

---
**2. On the hyperparameter analysis regarding freezing ratios** (Response to Reviewers t7Au, A7vK)

We conducted a fine-grained ablation study of the freezing ratio for our key baselines, HFT and GMT, to address concerns about fair comparison. The corresponding results are added to the existing Figure 2. We confirm that HFT fails to surpass SSU across tasks and freezing ratios. Furthermore, while GMT achieves strong target performance at ratios above 60%, it consistently yields lower performance on source tasks than SSU, regardless of the freezing ratio. This confirms SSU as the optimal method for simultaneously achieving strong source preservation and high target language gains.

---
**3. On the Sensitivity to Calibration Data Size** (Response to Reviewer t7Au)

To address concerns regarding unfair source data benefits for SSU, we analyzed the sensitivity of the method to the size of the calibration set. Specifically, we reduced the calibration set from the default 500 examples to just 128 examples (~0.26M tokens). Table 5 confirms that a small sample set suffices for effective importance scoring.

Furthermore, Table 3 clearly shows that naive element-wise freezing disrupts feature transformations and causes catastrophic forgetting. This suggests that simply using source-related data does not retain the source capabilities of the model.

Consequently, we attribute the effectiveness of SSU to the structural (column-wise) protection mechanism, rather than the quantity of calibration data.

---
**4. On the textual revisions**

We have also made the following textual improvements to the paper:

**Related work**:
* **Comparison against Continual Learning (CL)** (Footnote 3 & Appendix E) (Response to Reviewer 7cj2): We extended the related work discussion to situate SSU within the parameter-centric family of CL solutions.
* **Mitigation of catastrophic forgetting for Multilingual CPT** (Section 2: L119-125) (Response to Reviewer t7Au): We added a clear discussion of the existing literature on methods for mitigating catastrophic forgetting in multilingual CPT.
* **Comparison against importance methods in different paradigms** (L143-148) (Response to Reviewer t7Au): We included a discussion on related methods that utilize importance-based techniques in contexts differing from ours.

**Additional clarifications**:
* **Refined claims in L56-59** (Response to Reviewer 7cj2): We addressed feedback regarding the breadth of the original claim about "existing paradigms".
* **Tokenization issues (Footnote 2)** (Response to Reviewer 7cj2): We clarified that over-fragmentation issues in underrepresented languages are important but orthogonal to our setting.

---
We believe these revisions comprehensively address your feedback and demonstrate the validity of SSU.

We are happy to answer any further questions.

Sincerely,
The Authors

---

### Author Response · Authors · 2025-11-25
**Request for Feedback on Revision & Response**

Dear Reviewers gbcc, 7cj2, t7Au, and A7vK,

We hope this message finds you well.

We recently submitted our detailed author response and revised manuscript to address the valuable feedback received during the initial review phase.

We understand your time is extremely limited, but we would be very grateful if you could briefly confirm whether our rebuttal and the corresponding changes in the revised manuscript adequately addressed your major concerns.

Thank you once again for your review.

Sincerely,

The Authors of Submission 9371

---

### Note · Authors · 2025-12-04

**Comment:**

We sincerely thank the anonymous AC and reviewers for their constructive feedback.

Following careful consideration of the reviews, and acknowledging the very limited opportunity for discussion due to the OpenReview technical incident, we have decided to withdraw the current submission.

We plan to submit a revised and updated version of the paper to an alternative venue in the future.

**Withdrawal Confirmation:**

I have read and agree with the venue's withdrawal policy on behalf of myself and my co-authors.